# *In Vitro* Modelling of Oral Microbial Invasion in the Human Colon

Lucie Etienne-Mesmin,[a] Victoria Meslier,[b] Ophélie Uriot,[a] Elora Fournier,[a] Charlotte Deschamps,[a] Sylvain Denis,[a] Aymeric David,[b] Sarah Jegou,[b] Christian Morabito,[b] Benoit Quinquis,[b] Florence Thirion,[b] Florian Plaza Oñate,[b] Emmanuelle Le Chatelier,[b] S. Dusko Ehrlich,[b] Stéphanie Blanquet-Diot,[a] Mathieu Almeida[b]

aUMR 454 UCA-INRAE Microbiologie Environnement DIgestif et Santé (MEDIS), Université Clermont Auvergne, Clermont-Ferrand, France
bUniversité Paris-Saclay, INRAE, MetaGenoPolis (MGP), Jouy-en-Josas, France

Lucie Etienne-Mesmin and Victoria Meslier contributed equally to this work. Author order was determined alphabetically.

**ABSTRACT** Recent advances in the human microbiome characterization have revealed significant oral microbial detection in stools of dysbiotic patients. However, little is known about the potential interactions of these invasive oral microorganisms with commensal intestinal microbiota and the host. In this proof-of-concept study, we proposed a new model of oral-to-gut invasion by the combined use of an *in vitro* model simulating both the physicochemical and microbial (lumen- and mucus-associated microbes) parameters of the human colon (M-ARCOL), a salivary enrichment protocol, and whole-metagenome shotgun sequencing. Oral invasion of the intestinal microbiota was simulated by injection of enriched saliva in the *in vitro* colon model inoculated with a fecal sample from the same healthy adult donor. The mucosal compartment of M-ARCOL was able to retain the highest species richness levels over time, while species richness levels decreased in the luminal compartment. This study also showed that oral microorganisms preferably colonized the mucosal microenvironment, suggesting potential oral-to-intestinal mucosal competitions. This new model of oral-to-gut invasion can provide useful mechanistic insights into the role of oral microbiome in various disease processes.

**IMPORTANCE** Here, we propose a new model of oral-to-gut invasion by the combined use of an *in vitro* model simulating both the physicochemical and microbial (lumen- and mucus-associated microbes) parameters of the human colon (M-ARCOL), a salivary enrichment protocol, and whole-metagenome shotgun sequencing. Our study revealed the importance of integrating the mucus compartment, which retained higher microbial richness during fermentation, showed the preference of oral microbial invaders for the mucosal resources, and indicated potential oral-to-intestinal mucosal competitions. It also underlined promising opportunities to further understand mechanisms of oral invasion into the human gut microbiome, define microbe-microbe and mucus-microbe interactions in a compartmentalized fashion, and help to better characterize the potential of oral microbial invasion and their persistence in the gut.

**KEYWORDS** oral microbial invasion, gut microbiota, mucus, M-ARCOL, metagenomics, oral microbiota

The human gastrointestinal (GI) tract harbors a vast and complex community including between 10 trillion and 100 trillion microbes dominated by bacteria, collectively referred to as the gut microbiota (1). The gut microbiota plays a major role in host physiology, with an involvement in energy extraction from food, vitamin synthesis, maturation of the immune system, and protection against invasion by enteric pathogens (2, 3). In the human body, oral and GI microbiomes represent the two largest microbial ecosystems (4, 5), and pioneering data from the Human Microbiome

Address correspondence to Mathieu Almeida, mathieu.almeida@inrae.fr, or Stéphanie Blanquet-Diot, stephanie.blanquet@uca.fr.

The authors declare no conflict of interest.

10.1128/spectrum.04344-22 **1**

Project demonstrated that they are taxonomically diverse, representing 26% and 29% of total bacteria from the human body, respectively (6).

Studies have shown that saliva contains approximately $10^7$ to $10^9$ bacteria per milliliter (7, 8), with a global diversity of approximately 700 species listed from the oral cavity of healthy subjects. These species are members of *Firmicutes*, *Proteobacteria*, *Bacteroidetes*, *Actinobacteria*, *Fusobacteria*, *Spirochaetes*, *Synergistetes*, and TM7 (5, 9). *Streptococcus* is the most abundant genus in the oral site, with *Haemophilus*, *Veillonella*, and *Prevotella* also prevalent (5, 10, 11). Countless studies have demonstrated that microbiota composition distinctively changes all along the GI tract due to differences in term of oxygenation, substrate availability, pH, and residence time between digestive compartments, therefore inducing microbial species niche preferences (12). The highest bacterial load ($10^{11}$ to $10^{12}$ bacteria per g) and diversity are reached in the colon, where predominant phyla are *Bacteroidetes*, *Firmicutes*, *Actinobacteria*, and, to a lesser extent, *Proteobacteria* and *Verrucomicrobia* (13–16). It is now well established that each individual harbors a unique gut microbiota composed of an estimated 300 bacterial species detected per healthy individual on average (17–19).

Despite physical distance and chemical hurdles that keep apart the oral microbiome from the gut microbiome, cumulative evidence supports the notion that the oral microbiota is present in the overall gut microbial ecosystem. Li et al. demonstrated that transplantation of human saliva to gnotobiotic mice led to a distribution of oral genera throughout the GI tract, with *Fusobacterium*, *Haemophilus*, *Streptococcus*, and *Veillonella* being especially abundant in the gut of recipient mice (20). In humans, independent studies have demonstrated that some bacterial genera detected in the same healthy subject can overlap between oral and stool samples, confirming an extensive transmission of microbes through the GI tract (11, 21). Such a phenomenon of oral-gut transmission occurring under physiological conditions seems to be amplified in a pathological context. Orally derived microorganisms are particularly enriched in patients with altered gut microbiota (perturbation termed dysbiosis) and barrier disruption. In particular, Hu and colleagues showed that oral bacteria are enriched in the fecal microbiota of Crohn's disease patients (22), suggesting that the oral cavity might act as a reservoir of opportunistic pathogens with the ability to colonize the gut (23), even more important in such susceptible hosts. Likewise, a large fraction of species enriched in the fecal microbiota of patients with liver cirrhosis or after bariatric surgery are of oral origin (24, 25).

To date, the interconnections between oral and gut microbiota have not been fully elucidated and mechanisms associated with the gut colonization by oral bacteria are not clear. This can be explained by (i) the technical difficulties met when analyzing oral microbial samples with high-resolution shotgun metagenomic sequencing, due to the high proportions of retrieved host DNA, and (ii) the lack of relevant models. Clinical trials remain the gold standard approaches but are hampered by heavy regulatory, technical, and costly constraints. For evident ethical reasons, human gut microbiota studies are usually performed with fecal samples, making result interpretation difficult since direct access to the different segments of the GI tract—especially the colon—is limited. Animal models integrate host-microbe interactions, but translation to the human situation remains limited due to major differences of digestive physiology and the oral and gut microbiotas between most animal models and humans.

A relevant alternative in preclinical studies involves the use of an *in vitro* model simulating the human digestive environment. *In vitro* models permit accurate reproduction of the complexity and diversity of the *in vivo* microbial ecosystem (26–28) and were recently optimized to incorporate mucin beads leading to mucosal configuration of the models, i.e., mucosal simulator of the human intestinal microbial ecosystem (M-SHIME) and Mucosal ARtificial COLon (M-ARCOL) (13, 28–32). In the present study, we investigated oral-to-gut microbial invasion by using the M-ARCOL. Our main goal was to validate our experimental approach using shotgun metagenomic analysis of stools

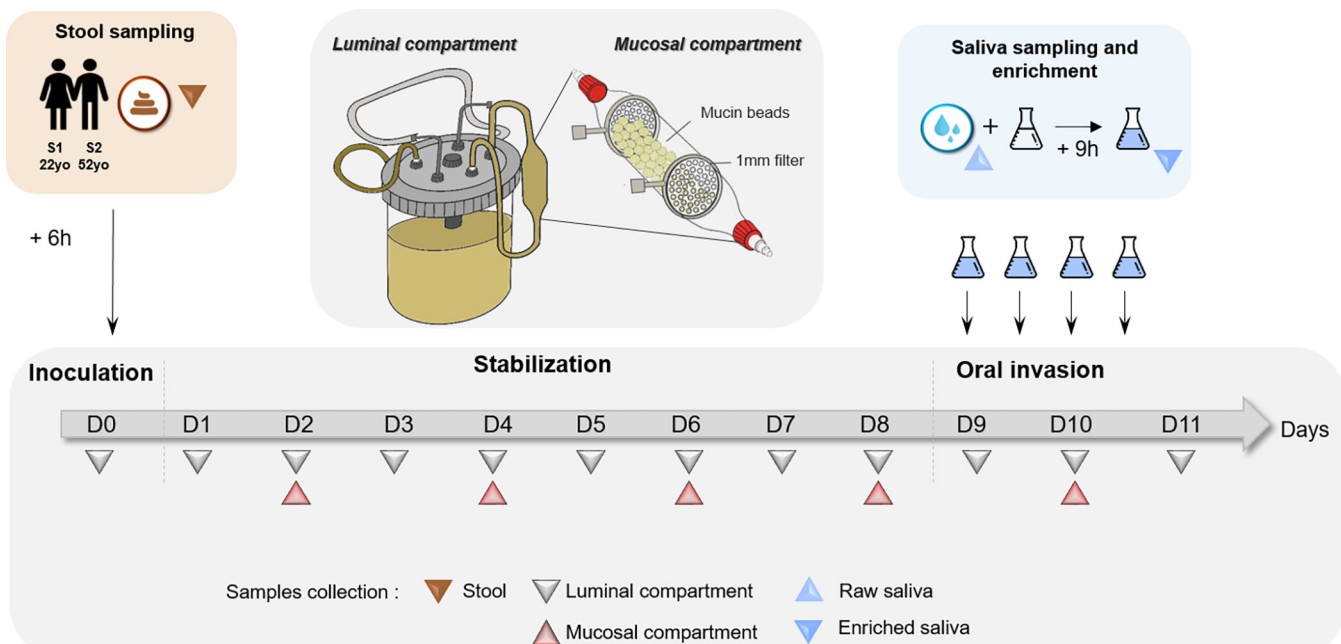

**FIG 1** Experimental workflow for *in vitro* fermentation setup and oral invasion. Fresh stool samples from two healthy donors (S1 and S2) were used to inoculate two independent M-ARCOL bioreactors. Each fermentation was conducted for a total period of 11 days, including 24 h of batch amplification and 10 days of continuous fermentation. After an 8-day stabilization period, oral-to-gut invasion was simulated by injecting a 9-h enriched saliva from the same donor, twice a day and for 2 consecutive days (morning and late afternoon of days 9 and 10). Samples were collected from fresh stools (brown triangle), the luminal compartment of the bioreactor every day (gray triangle), and the mucosal compartment every 2 days (red triangle), as were raw saliva and enriched saliva samples (blue triangle) for each donor.

and saliva samples from healthy donors and to assess the effects of oral microbial invasion on the luminal and mucosal microenvironments in the simulated human colon.

## RESULTS

**Development of a novel experimental design for oral-to-gut invasion in M-ARCOL bioreactors.** In this study, we developed an oral-to-gut invasion model using a one-stage fermentation system (M-ARCOL) setup to reproduce the main physicochemical parameters (pH, temperature, transit time, nutrient availability) found in the human colon (Fig. 1). M-ARCOL bioreactors, composed of two compartments used to mimic the luminal and mucosal microenvironments, were inoculated with fecal samples from two healthy adults. This study was conducted on fecal samples collected from two healthy donors, chosen to represent "extreme" conditions based on their sex, age, and nutritional habits: donor S1 is a methane producer female (age, 22 years; flexitarian-based (neo vegetarian) diet), while donor S2 is a non-methane producer male (age, 52 years; omnivorous diet). Gas analysis confirmed that anaerobic conditions were efficiently maintained in the bioreactors by the sole metabolic activity of the gut microbiota without gas flushing, which constitutes a main feature of the M-ARCOL model compared to other colonic *in vitro* models (see Fig. S1A and B in the supplemental material). Notably, methane production was detected for one of the donors, indicating the presence of methanogenic microorganisms. The three main short-chain fatty acids (SCFAs; acetate, butyrate, and propionate) were also measured in the luminal compartment with ratio similar to that found *in vivo* in the human colon, as previously validated (28) (Fig. S1B and C). To allow a combined oral invasion experiment and shotgun sequencing on human saliva samples, we generated enriched microbial saliva samples and inoculated them at days 9 and 10 in the luminal phase of M-ARCOL to evaluate oral-to-gut microbial invasion after gut microbial stabilization in the bioreactors (Fig. 1). DNA was extracted from all collected samples and subjected to shotgun metagenomic sequencing. All samples from fecal origins (raw stool and luminal and mucosal samples from the M-ARCOL) displayed high mapping rates onto the IGC2 gut gene

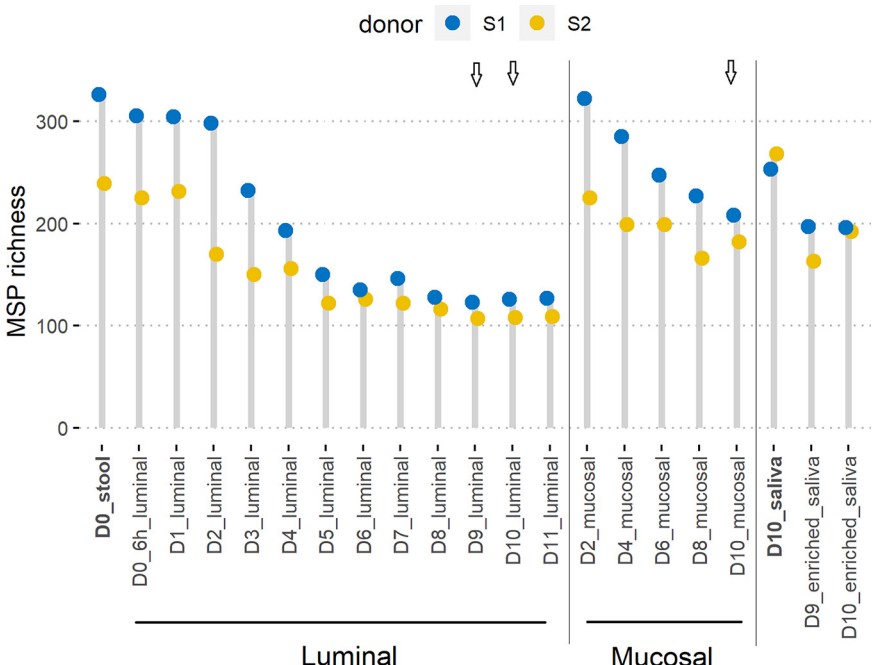

**FIG 2** Dot chart for MSP richness dynamic over time. Metagenomic species pangenome (MSP) species richness was calculated as the number of detected MSP species in the corresponding sample for donors S1 and S2 on the merged MSP abundance table, for luminal and mucosal compartments of the M-ARCOL and for raw saliva and enriched-saliva samples. In bold are indicated initial raw stool and saliva samples. Times of fermentation in the colonic M-ARCOL model are indicated in days (D). Arrows indicate saliva injection into the bioreactors on days 9 and 10.

catalogue (median rates > 80%) but not onto the oral catalogue (median near 5%). Salivary samples displayed high mapping rates onto the oral microbial gene catalogue (median > 79%) and less than 40% mapping rates onto the gut microbial gene catalogue (Fig. S2 and Table S1). Profiles of atmospheric gases and luminal SCFAs were not modified by addition of enriched saliva (Fig. S1).

**The mucosal microenvironment retained higher microbial richness during *in vitro* fermentation.** We determined the metagenomic species pangenome (MSP) richness, defined as clusters of coabundant genes and representative of microbial species, over time and in the different compartments of the M-ARCOL (Fig. 2). For both donors, initial stool and raw saliva samples displayed the highest MSP species richness compared to the bioreactor samples. During *in vitro* fermentations, we observed a loss of richness in the luminal and mucosal compartments, until a stabilization at day 5 in the luminal compartment for both donors. Consistent with fecal MSP richness, donor S1 displayed a higher MSP richness during the first days of fermentation in the luminal and mucosal compartments than did donor S2 (delta of 87 and 15 MSP between donors S1 and S2 for stool and raw saliva, respectively). After MSP species richness equilibrium and until the end of the experiment, MSP richness levels were almost equivalent between the two donors in the luminal compartment (delta of 18 MSP at day 11), and individual microbial signature was maintained, as estimated from Bray-Curtis distance measures (Fig. S3). In the mucosal compartment, MSP richness was found to be systematically higher than the luminal one at each time point for both donors. The richness loss observed was thus lesser and slower all along the process. The MSP stool richness difference between the donors persisted longer (until at least day 8) in the mucosal compartment than in the luminal one (delta of 12 MSP in the luminal and 61 MSP in the mucosal compartment for donors S1 and S2 at day 8). The MSP species richnesses were found to be equivalent between donors S1 and S2 in raw saliva samples (delta of 15 MSP), and they decreased in enriched saliva of both donors at days 9 and 10 (delta of 4 MSP between donors S1 and S2 at day 10).

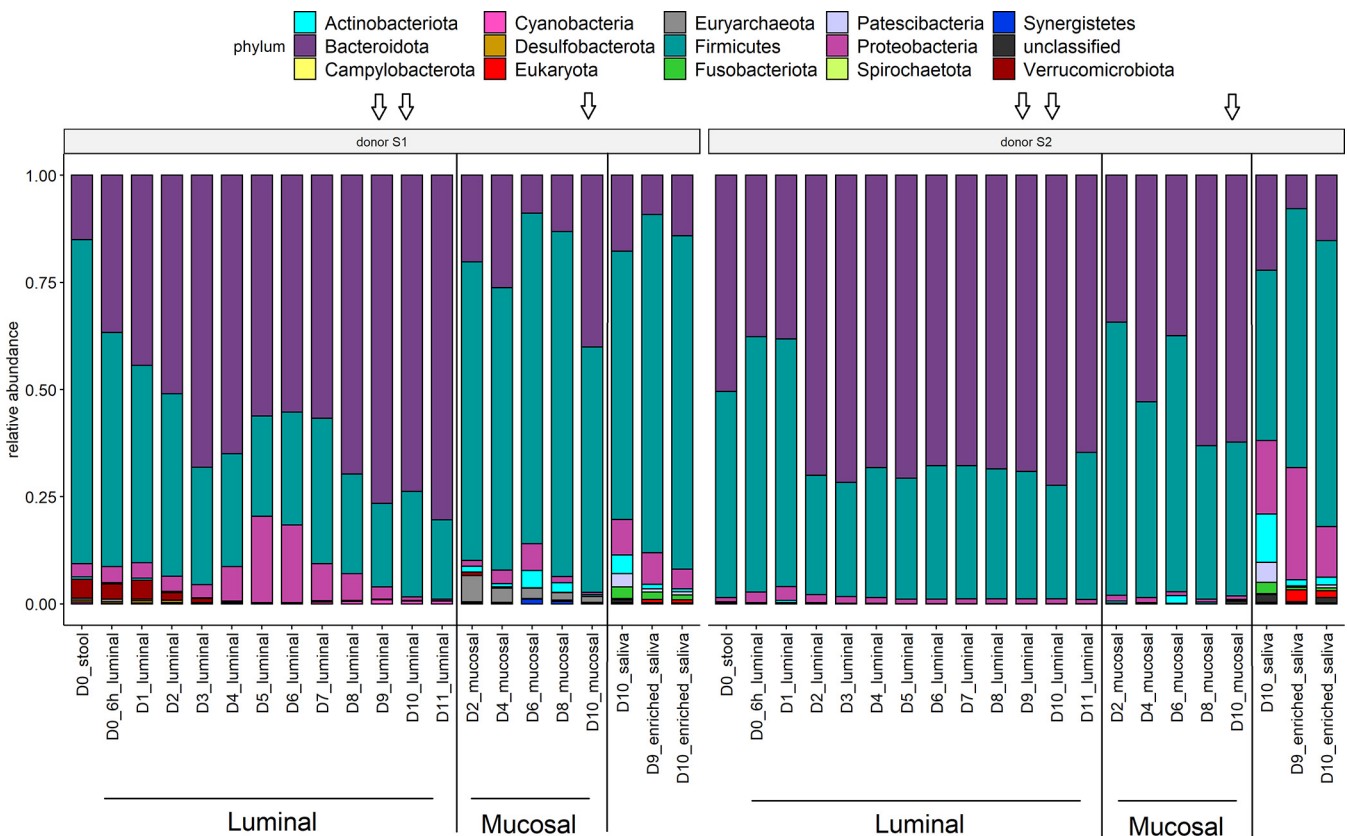

**FIG 3** Phylum rank normalized composition per donor. MSP species abundance was normalized per sample by dividing its abundance by the sum of the MSP species abundances detected in the sample. Phylum rank composition was calculated as the sum of the normalized abundances of the corresponding MSP species. Donors (S1 or S2), M-ARCOL compartments (luminal and mucosal), and the days of fermentation are reported. Arrows indicate saliva injection into the bioreactors on days 9 and 10.

**Differential microbial compositions between stool, salivary, luminal, and mucosal samples.** Based on Bray-Curtis dissimilarity distance, the overall stool, luminal, and mucosal compositions appeared quite close for a given individual and contrasted between the two donors, which was also observed in the salivary microbiota (Fig. S3). Indeed, we found that samples clustered together by donor and sample type but also by time points. After the injection of saliva into the bioreactors on days 9 and 10, the luminal and mucosal samples collected were not distant from the same-donor fecal samples, suggesting that dominant microbial compositions were not drastically modified by the oral microbial administration. We also observed that raw saliva and enriched saliva clustered together for each donor, confirming their close microbial compositions.

We analyzed the composition of saliva, luminal, and mucosal microbial communities in the M-ARCOL at the phylum (Fig. 3) and family (Fig. S4) ranks. At the phylum rank, we found a dominance of *Firmicutes* and *Bacteroidetes* followed by *Proteobacteria*, with some differences between the two donors or between compartments. Main families included *Bacteroidaceae*, *Rikenellaceae*, and *Prevotellaceae* (*Bacteroidetes*) across samples and *Ruminococcaceae* and *Streptococcaceae* (*Firmicutes*) for fecal and oral samples, respectively. Stool of donor S1 was dominated by *Firmicutes* (0.75 relative abundance), and stool of donor S2 contained equivalent levels of *Bacteroidetes* and *Firmicutes* (0.5 and 0.47 relative abundances, respectively). While the ratio between *Firmicutes* and *Bacteroidetes* was modified in the luminal compartment over time, the primary ratio for these taxa observed in stool samples was globally maintained in mucosal samples, particularly for donor S2. An additional phylum was the *Verrucomicrobiota* phylum (including *Akkermansia muciniphila*), detected in the two donors in the luminal compartment but at higher levels in donor S1 (100 times more; maximum relative abundance of 0.04). The *Euryarchaeota*

phylum (*Methanobrevibacter smithii* species), detected exclusively in donor S1, consistent with methane detection in gas analyses (Fig. S1A), was found in low relative abundance in stool and the luminal compartment (relative abundance of <0.01) but was identified in higher levels in the mucosal samples already at day 2 (relative abundance between 0.01 and 0.06). Members of the *Actinobacteriota* phylum (mainly of the *Micrococcaceae*, *Bifidobacteriaceae*, and *Actinomycetaceae* families) were found in higher relative abundances in the mucosal and salivary samples (up to 0.02), were low in stool samples (below 0.01), and decreased for donor S1 after the oral-to-intestinal invasion. In salivary samples, ratios of *Firmicutes* to *Bacteroidetes* were found close to those observed in fecal samples for both donors, with additional phyla detected in these samples, such as higher levels of *Proteobacteria*, *Fusobacteria*, *Patescibacteria*, and *Actinobacteriota*. No major changes in raw or enriched saliva samples were found for either donor, as confirmed by Bray-Curtis dissimilarity analysis (Fig. S3).

**A few oral microbial species are present in the luminal and mucosal compartments of the M-ARCOL before oral invasion.** To evaluate the impact of the oral-to-gut invasion on the M-ARCOL microbial composition, we assigned the species to their preferred ecological niche, using their occurrence in raw stools or saliva samples, and three species types were defined as gut, oral, or not determined (ND; species that were either undetected in raw stool and raw saliva samples or detected in both stool and saliva samples before inoculation of the bioreactor at the initial time point [T0] [Table S2]). We then analyzed the niche distribution to assess the number of oral species before and after the invasion in the luminal and mucosal compartments (Fig. 4) and the relative abundance of each ecological niche (Fig. S5 and Table S3). Raw stools were exclusively dominated by gut-oriented MSP species and ND species, with no oral-oriented MSP species detected for either donor, and raw saliva was dominated exclusively by oral-oriented MSP species and ND species (Fig. 4, Fig. S5, and Table S3). Before oral-to-gut invasion, a dominance of oral-oriented species was observed in enriched saliva samples, yet four and seven ND MSP species were detected, respectively, in donors S1 and S2 (Fig. 4, Fig. S5, and Table S3).

Following stool inoculation of the bioreactor, luminal and mucosal compartments were largely dominated by gut MSP species, yet two oral MSP species were detected before simulated oral-to-gut invasion (Fig. 4 and Table S3). For donor S1, these oral MSP species were detected in luminal and mucosal samples (msp_0616 *Prevotella buccae* and msp_1193 *Dialister pneumosintes*), raising to up to a fourth of the relative abundance at day 8 in the luminal sample but remaining at low relative abundance (0.01) in the mucosal sample before oral-to-gut invasion. For donor S2, only one MSP species (msp_0616 *Prevotella buccae*) was detected in the luminal compartment (Fig. S5). Oral MSP species were nondominant, either by number or relative abundance, at day 8 in the mucosal compartment of the 2 donors before oral-to-gut invasion.

**Preference of oral microbial invaders for the mucosal microenvironment.** After the injection of enriched saliva in the system, we found only three oral MSP species in the luminal compartments of both donors at days 9 and 10 (msp_0616 *Prevotella buccae*, msp_0677c *Slackia exigua*, and msp_0884 *Veillonella atypica*) (Fig. 5); these low numbers persisted at day 11. In contrast, a much higher number ($n = 28$) was detected in the mucosal microenvironment of both donors, representing about 15% of the enriched saliva microbial diversity. Similar numbers of oral MSP species were found for donors S1 and S2 in the mucosal samples (17 and 20 oral MSP species, respectively), but their relative abundances differed between donors (relative abundances of 0.13 and 0.0043, respectively [Fig. S5]). These oral species invaders belonged to a limited number of taxa, with members of the genera *Veillonella*, *Streptococcus*, *Prevotella*, and *Haemophilus*, all common taxa of the oral microbiome (Fig. 5). Additional common taxa included *Porphyromonas*, *Neisseria* and *Rothia* genera. We also observed that the relative abundance of these 28 oral invaders in the mucosal microenvironment was not systematically related to their respective abundance in enriched saliva samples (rho Spearman, 0.21 and 0.88 for donors S1 and S2, respectively, between enriched

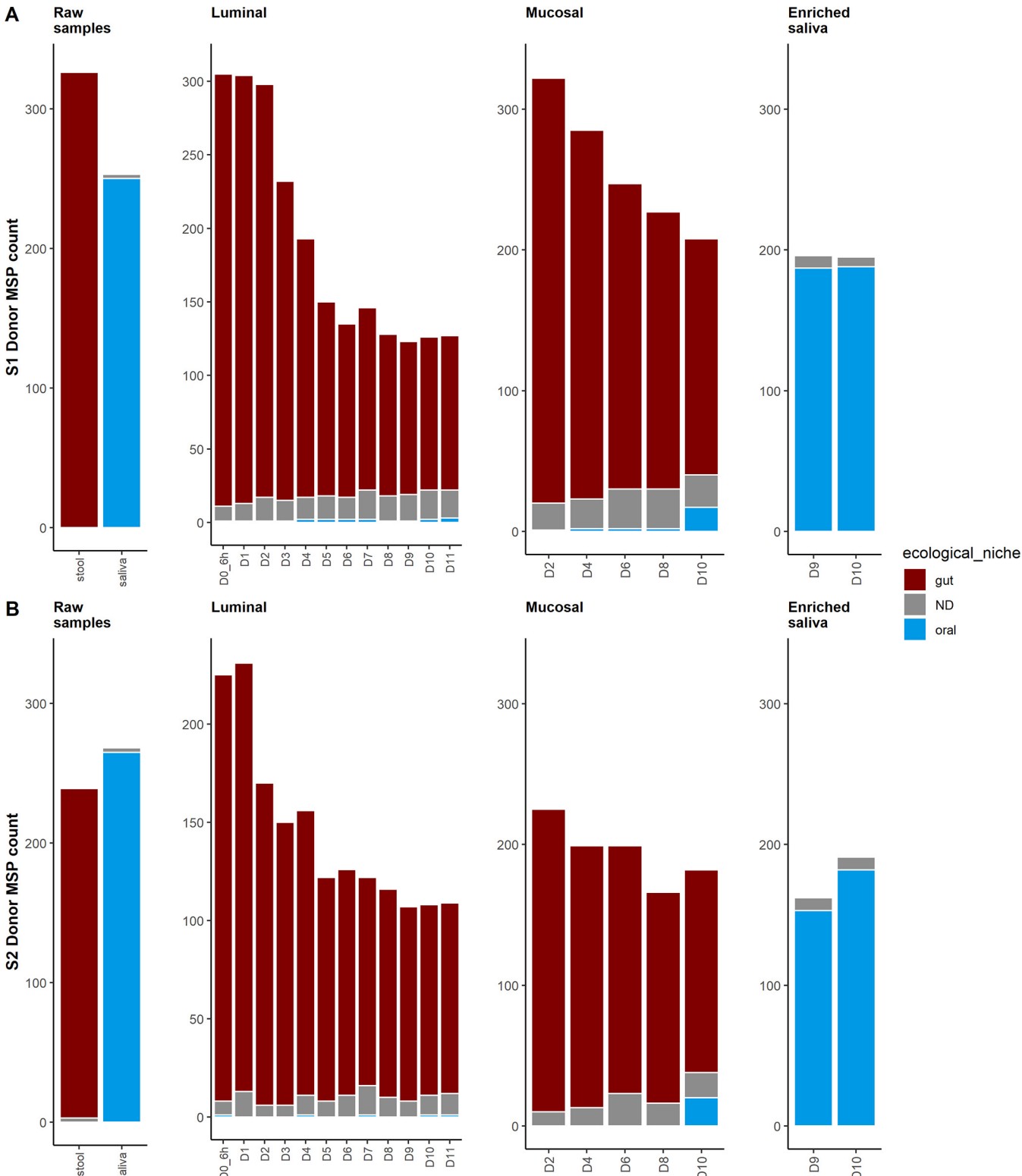

**FIG 4** MSP species richness of oral-to-gut invasion using species ecological niche. MSP species richness was split by their corresponding ecological niche, namely, the dominant ecological ecosystem. (A) donor S1; (B) donor S2. Times of fermentation in the colonic M-ARCOL model are indicated in days.

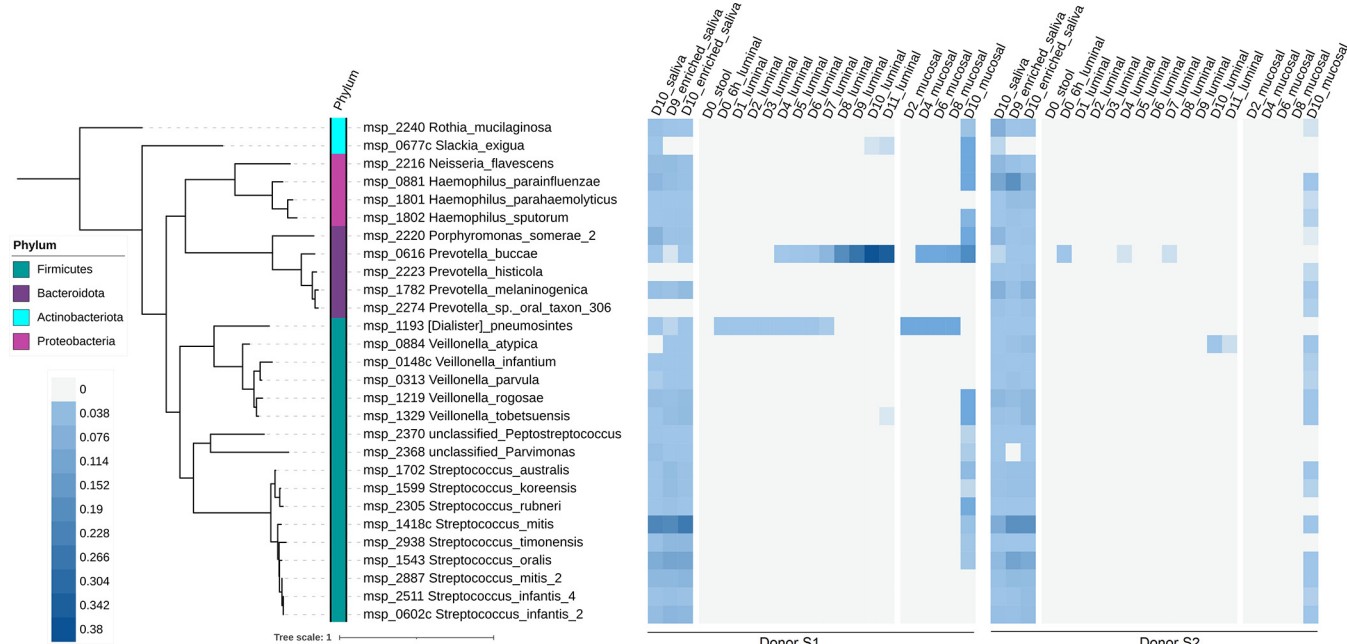

**FIG 5** Phylogenetic tree for oral MSP species during oral-to-gut invasion. Shown is the phylogenetic tree for 28 oral MSP species detected in the initial stools and the luminal and mucosal samples from M-ARCOL, regardless of the raw and enriched saliva composition. The tree was generated using the 40 universal marker proteins from the oral MSP species extracted by the fetchMG software. For each of donors S1 and S2, a heat map for the oral MSP species normalized abundance is shown, with a color shade from gray (not detected) to dark blue (highly detected; maximum relative abundance = 0.38). Times of fermentation in the colonic M-ARCOL model are indicated in days.

saliva and mucosal at day 10; *P* values = 0.28 and < 1.5e−9). At the end of the experiment (day 11), only three and one oral invader species were detected, respectively, for donors S1 and S2 in the luminal compartment (Fig. 5).

## DISCUSSION

Here, we present a new model for oral-to-gut microbial invasion simulation using the M-ARCOL bioreactor and whole-metagenome sequencing, with the goal to provide a tool for a comprehensive understanding of the interactions of these two microbial ecosystems of the gastrointestinal tract. This experimental setup was motivated by the growing body of evidence for the oral microbial communities' impact on the gut microbiome composition and its potential effect on human health (22, 24, 33, 34). The new model consists of a colonic bioreactor inoculated with fecal samples from healthy adult donors, used to follow daily microbial changes for 11 days, combined with a simulated oral invasion by injecting enriched saliva on days 9 to 10. A shotgun metagenomics-based analysis was essential to achieve species-level resolution, to differentiate closely related species originating from the oral or gut ecosystem. This study also required a combined salivary and fecal *in silico* microbial exploration of two healthy donors with distinct fecal microbial compositions: *Firmicutes* dominance for donor S1 and *Bacteroidetes* dominance for donor S2. Since fecal microbiota interindividual variability is important, this pilot study was carried out on fecal samples collected from two healthy donors known to represent different conditions in terms of fecal microbial composition, methane-producing status, age (S1, 22 years; S2, 52 years), and dietary habits (S1, flexitarian (neo vegetarian) diet; S2, omnivorous diet). Lastly, we assessed the oral-to-gut microbial interactions using the mucosal configuration of the ARCOL *in vitro* gut model (28), providing a unique opportunity to independently investigate lumen- and mucosa-associated microbial communities, thanks to a distinct capture of the fine-scale regionalization of the human colon (28).

One key aspect of our investigation was to retrieve enough salivary material to perform both the oral injection into the bioreactor and the microbial DNA extraction for

shotgun metagenomic analysis. To circumvent a relatively small amount of saliva biomass, we performed a microbial saliva enrichment step prior to oral injection. While this enrichment reduced somewhat the oral microbial richness, a major part was retained (70%), with a composition close to that of the raw saliva, thus preserving the oral microbial signatures of the two donors. We also observed that the human DNA was still detected after saliva enrichment. We hypothesize that human DNA remained during enrichment by the formation of microbial aggregates seeding upon human salivary mucosa and associated host cells, as recently observed by Simon-Soro and colleagues (35). Interestingly, we found that, whatever the donor, the mucosal compartment of the M-ARCOL system enabled the subsistence of a higher microbial richness than the luminal one (28, 36, 37). This highlights the critical contribution of the mucosal setup in microbial dynamic analysis based on colonic *in vitro* models and supports the underestimated role of the mucus in many physiological and pathological processes involving the gut microbiome (38–40).

Using the microbial species ecological niche predisposition (gut or oral), we explored the number and amount of oral microbial species in the luminal and mucosal compartments of the M-ARCOL before and after simulated oral invasion. As expected, variations of oral microbial species after invasion were detected, indicating that the protocol design was successful in emulating the two microbial ecosystems. While the two donors displayed different raw oral and fecal microbial compositions, it was observed that oral invasion mostly occurred in the mucosal compartment and was limited in the luminal one for both donors. This suggests invasive oral species preference for the intestinal mucosal resources and that abundance of oral invaders alone, but likely the species-specific phenotypes, permitted bacteria to invade the mucosa. Interestingly, the oral invaders were all members of common salivary microbiota, as part of the healthy oral core microbiome (41–44). These results suggest that some oral species might possess functions to utilize MUC-2 proteins from the gut (45). We also observed for donor S1 a significant increase and resilience of several oral species prior to the simulated oral invasion, indicating that subabundant oral species present in stool samples could surge when appropriate conditions occurred, such as a decrease in gut microbial richness (19, 46).

The main objectives of the current study were to test the feasibility of the experimental approach to simulate oral microbial invasion in the intestinal compartment and the efficiency of the saliva enrichment. Our study demonstrates promising opportunities to (i) further understand mechanisms of oral microbial invasion of the human gut microbiome (5, 47), (ii) define microbe-microbe interactions in a compartmentalized fashion (luminal versus mucosal), and (iii) eventually help to clarify the potential impact of oral invasion on human health (20, 48). Future developments could include the implementation of an upper *in vitro* human digestive tract by coupling the M-ARCOL bioreactor with the TIM-1 stomach and small intestinal digester (30, 49, 50). This would allow the combined determination of oral microbiota survival and oral-to-intestinal microbial interactions. Moreover, it would be needed to study the invasion of oral microbes in the *in vitro* M-ARCOL model for a longer period to shed new light on long-term invasion outcomes. Several studies have shown that orally derived bacteria can colonize and persist better within the gut under diseased conditions (5, 23, 24, 33). Colon *in vitro* gut models, including the M-ARCOL, can be advantageously adapted to mimic pathological situations associated with gut microbial dysbiosis, such as obesity (51, 52), irritable bowel syndrome (53), and inflammatory bowel disease (54, 55). This can be performed by inoculating them with fecal samples from patients but also adapting all the nutritional and physicochemical parameters to the diseased situations (28, 50). Understanding the correlation of the oral-gut microbiome axis in pathogenesis confers an advantage for precise diagnosis and effective treatment via targeted microbial strategies such as probiotics or fecal microbiota transplantation. In order to get closer to the *in vivo* situation by integrating host-microbiota interactions, further studies will be needed coupling *in vitro* colon models to intestinal epithelial cells or

immune cells, intestinal organoids, or bioengineered human gut-on-chip devices such as HuMiX, Intestine-Chip, and Colon-Chip systems (31, 50).

## MATERIALS AND METHODS

**Sample collection for bioreactor inoculation and oral invasion.** Donors (S1 and S2) were selected according to their sex (one female and one male), their age (23 and 52 years old), their dietary habits (one eating a flexitarian-based (neo vegetarian) diet and one consuming a omnivorous diet), and their methane status (one methane producer and one non-methane producer). The selected donors had no history of antibiotic treatment or drug or probiotic consumption for 3 months prior to sample collection. This study was a noninterventional study with no additions to usual clinical care. Fecal samples were prepared anaerobically as previously described (28), within 6 h after defecation. Saliva was collected from the same donor; each donor abstained from food and drink intake for 2 h prior to sample collection. Saliva was collected twice a day (morning and afternoon) by passive drool collection (no spitting and no blowing). Briefly, after allowing saliva to pool at the bottom of the mouth, the head was tilted forward, enabling a passive flow collected (10 mL) in a sterile container.

**Fermentation in M-ARCOL.** The Mucosal ARtificial COLon (M-ARCOL) is a one-stage fermentation system, used under continuous conditions, which simulates the physicochemical conditions encountered in the human colon as well as the lumen- and mucus-associated human intestinal microbial ecosystem (Applikon, Schiedam, The Netherlands) (28). It consists of pH-controlled, stirred, airtight glass vessels kept under anaerobic conditions maintained by the sole activity of resident microbiota, one vessel (300 mL) mimicking the lumen-associated microbiota and a second vessel containing mucin-alginate beads to mimic the mucus-associated microbiota (28). The vessels were operated with an initial sparging with $O_2$-free $N_2$ gas and then inoculated with fecal material from donors S1 and S2, respectively. This dynamic *in vitro* model was set up to reproduce conditions of a healthy human adult colon with a fixed temperature of 37°C, a controlled constant pH of 6.3, a stirring speed at 400 rpm, a mean retention time of 24 h, and a redox potential (Eh) of −200 mV. A sterile nutritive medium containing various sources of carbohydrates, proteins, lipids, minerals, and vitamins was sequentially introduced into the bioreactor as described previously (28, 56).

**Gas and Short Chain Fatty Acids (SCFA) analysis.** Analysis of $O_2$, $CO_2$, $CH_4$, and $H_2$ produced during the fermentation process in the atmospheric phase of the bioreactors was performed daily to ascertain that anaerobic conditions and gas composition were verified (Fig. S1). Gas composition was analyzed using an HP 6890 gas chromatograph (Agilent Technologies, USA) coupled with a Thermal Conductivity Detector (TCD) detector (Agilent Technologies). The three major SCFAs (acetate, butyrate, and propionate) were quantified in colonic samples from the luminal phase by high-performance liquid chromatography (HPLC) (Elite LaChrom, Merck Hitachi, USA) coupled with a Diode Array Detector (DAD) diode as described previously (28).

**Mucin beads and mucin compartment.** Type II mucin from porcine stomach (Sigma-Aldrich, USA) and sodium alginate (Sigma-Aldrich, USA) were diluted in sterile distilled water at concentrations of 5% and 2%, respectively. To produce mucin beads, the mucin-alginate solution was dropped into a 0.2 M solution of sterile $CaCl_2$ under agitation using a peristaltic pump. Mucin-alginate beads were introduced in the airtight glass compartment (total area of beads, 556 cm²) connected to the main bioreactor, allowing a continuous flow of the luminal medium and ensuring the contact of the resident luminal microbiota with the mucin beads. The mucin bead compartment was kept at 37°C through the experiment using a hot water bath. Mucin-alginate beads were replaced every 48 h with fresh sterile ones under a constant flow of $CO_2$ to retain anaerobiosis.

**Experimental design and sampling.** Following fecal inoculation of the bioreactor, fermentation was conducted for a total duration of 11 days. On days 9 and 10, 10 mL of enriched saliva was introduced into the bioreactor twice a day (morning and late afternoon) after an enrichment of 9 h. A saliva washout was realized on day 11 without the injection of enriched saliva samples. For saliva enrichment, 10 mL of freshly collected saliva was resuspended in 15 mL of colonic nutritive medium (37°C, anaerobically, 100 rpm) (28) to favor multiplication of oral bacteria and enrich the microbial fraction over the human fraction in the saliva samples. Circulating medium from the bioreactor vessel, referred to as luminal microbiota, was collected daily. Medium circulating in the mucin bead compartment, referred to as mucosal microbiota, was collected every 2 days. The remaining enriched saliva samples (∼5 mL) were collected for each donor (referred to as enriched saliva) to characterize enriched saliva. On day 10, a saliva sampling was performed for each donor without enrichment (referred to as raw saliva).

**DNA extraction procedures.** Prior to extraction, all samples were handled in the laboratory following International Human Microbiome Standards (IHMS) standard operating procedure (SOP) 004 (http://www.human-microbiome.org/) and stored at −80°C. Fecal and luminal samples were aliquoted into 200 mg and DNA extraction was performed following IHMS SOP P7 V2, which is adapted for low-microbial-biomass samples. A specific DNA preparation to remove human-related DNA was performed on saliva and enriched saliva as follow: 200 $\mu$L of saliva was centrifuged for 10 min at 10,000 × *g* for 4 min before adding 190 mL of sterile Milli-Q water to the pellet. After 5 min of incubation at room temperature, 10 $\mu$L of propidium monoazide (Clinisciences, Nanterre, France) was added to the tube to a final concentration of 10 $\mu$M. After 5 min of incubation at room temperature, samples were kept on ice and were exposed to halogen light for 25 min. Samples were stored at −80°C until extraction following IHMS SOP P7 V2. For mucosal samples, the same preparatory procedure was performed followed by DNA extraction using the IHMS SOP 06 V2 protocol, which is adapted to high-microbial-biomass samples. DNA was quantified using Qubit fluorometric

quantitation (Thermo Fisher Scientific, USA) and qualified using DNA size profiling on a fragment analyzer (Agilent Technologies, USA).

**High-throughput sequencing.** For each sample, 1 $\mu$g of high-molecular-weight DNA (>10 kbp) was used to build the library. DNA was sheared into 150-bp fragments using an ultrasonicator (Covaris, Woburn, MA), and DNA fragment library construction was performed using the Ion Plus fragment library and Ion Xpress barcode adapters kit (Thermo Fisher Scientific, USA), as previously described (57). We used an Ion Proton sequencer and Ion GeneStudio S5 prime sequencer to sequence the libraries (Thermo Fisher Scientific, USA), with a minimum of 20 million 150-bp high-quality reads generated per library for luminal, mucosal, and enriched saliva samples.

**Read mapping.** An average of 23.7 million ± 4.5 million reads was produced. We performed quality filtering using AlienTrimmer software to discard low-quality reads. Remaining human-related reads (0.02% for stools and luminal and mucosal samples and 61% for saliva and enriched saliva, on average) were removed using Bowtie2 (58), with at least 90% identity with human genome reference GRCh38 (GenBank assembly accession number GCA_000001405.15). Resulting high-quality reads were mapped onto the 10.4 million gut IGC2 (Integrated Gene Catalogue) catalogue of the human microbiome (59) and onto the 8.4 million oral catalogue (60) using METEOR software (https://forgemia.inra.fr/metagenopolis/meteor/-/tree/master/meteor-pipeline). Mapping was performed using a threshold of 90% for identity to the reference gene catalogue with Bowtie2 in a two-step procedure using a downsizing level of 12 million reads per sample, as described by Meslier et al. (61).

**MSP species determination and ecological niche definition.** The metagenomic species pangenome (MSP) was used to identify and quantify microbial species associated with the 10.4 million human gut genes on one hand (62) and the 8.4 million human oral genes on the other hand (60). MSP species are clusters of coabundant genes (minimum cluster size > 100 genes) used as a proxy for microbial species (63, 64). MSP abundances were estimated as the mean abundance of their 100 marker genes in both gut and oral catalogues, as far as at least 10% of these genes were detected (abundance strictly positive). From the independent gut and oral abundance tables, we computed a single abundance table by filtering overlapping MSP species between the two tables. MSP species ecological niche was determined by evaluating MSP detection in stools and raw saliva samples from the two donors. We assigned each MSP to gut and oral ecological niches when strict occurrence was found in either gut or saliva and not determined (ND) for MSP species that were either undetected in raw stool and raw saliva samples or detected in both stool and saliva samples before inoculation of the bioreactor at the initial time point (T0) (Table S2). MSP species richness was determined by counting the number of MSP species detected in the corresponding sample on the merged abundance table.

**Computational analysis.** All further steps were performed using R 3.5.0 (https://www.r-project.org). Data were processed and visualized using R packages *dplyr*, *stringr*, *tidyverse*, *ggpubr*, and *pheatmap*.

**Ethics declarations.** This study was a noninterventional study with no additions to usual clinical care. The donors provided written consent for the analysis and publication of the findings for their oral and fecal samples in the specific context of this study. This noninterventional study did not require approval from an ethics committee according to the French public health law (Code de la santé publique article L 1121-1.1).

**Data availability.** All sequencing data have been deposited at the European Nucleotide Archive database under the study accession PRJEB52431. Associated metadata are provided in Table S1.

## SUPPLEMENTAL MATERIAL

Supplemental material is available online only.

**SUPPLEMENTAL FILE 1**, XLSX file, 0.02 MB.
**SUPPLEMENTAL FILE 2**, XLSX file, 0.1 MB.
**SUPPLEMENTAL FILE 3**, XLSX file, 0.02 MB.
**SUPPLEMENTAL FILE 4**, PDF file, 0.7 MB.

## ACKNOWLEDGMENTS

This work was supported by the European FP7 Marie Skłodowska-Curie actions AgreenSkillsPlus PCOFUND-GA-2013-609398 grant to M.A. Additional funding was from Metagenopolis grant ANR-11-DPBS-0001 and from the SCUSI OBFIBRE program from Auvergne Rhône Alpes Region to MEDIS.

Oral invasion experiment design, M.A. and S.D.E.; M-ARCOL experiment design, S.B.-D. and L.E.-M.; bioreactor monitoring and sampling, L.E.-M., O.U., C.D., E.F., and S.D.; DNA extraction, A.D., S.J., and C.M.; metagenomic sequencing, B.Q.; metagenomic data preprocessing, V.M. and F.T.; oral microbial catalogue providers, F.P.O. and E.L.C.; data analysis and figures, V.M. and M.A.; data interpretation, V.M., M.A., L.E.-M., and S.B.-D.; writing, V.M., L.E.-M., M.A., and S.B.-D.; review and editing, V.M., L.E.-M., M.A., and S.B.-D.; resources, M.A. and S.B.-D.

We declare no conflicts of interest.

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
