## [Reviewer comments · Microbiology Spectrum]

Microbiology Spectrum

In vitro modelling of oral microbial invasion in the human colon

Lucie Etienne Mesmin, Victoria Meslier, Ophélie Uriot, Elora Fournier, Charlotte Deschamps, Sylvain Denis, Aymeric David, Sarah Jegou, Christian Morabito, Benoît Quinquis, Florence Thirion, Florian Plaza Oñate, Emmanuelle Le Chatelier, Stanislav Ehrlich, Stéphanie Blanquet-Diot, and Mathieu Almeida

Corresponding Authors: Mathieu Almeida, Université Paris-Saclay, INRAE, and Stéphanie Blanquet-Diot, University of Clermont-Auvergne, UMR UCA-INRA 454 MEDIS

Review Timeline:

Submission Date:	October 24, 2022
Editorial Decision:	December 29, 2022
Revision Received:	February 12, 2023
Accepted:	March 5, 2023

Editor: Diyan Li

Reviewer(s): The reviewers have opted to remain anonymous.

Transaction Report:

DOI: <https://doi.org/10.1128/spectrum.04344-22>

December 29, 2022

Dr. Mathieu Almeida
Université Paris-Saclay, INRAE
US 1367 MetaGenoPolis (MGP)
Saclay
France

Re: Spectrum04344-22 (In vitro modelling of oral microbial invasion in the human colon)

Dear Dr. Mathieu Almeida:

Link Not Available

Sincerely,

Diyang Li

Journals Department
Reviewer comments:

Reviewer #1 (Public repository details (Required)):

This study include sequencing data

Reviewer #1 (Comments for the Author):

The authors propose a new model of oral to gut invasion by the combined use of an in vitro model. This is a very interesting work. However, couples of concerns must be addressed by the authors before the considerable publication.

1 The authors claimed that their approach could provide useful mechanistic insights into the role of the oral microbiome in various disease processes. Therefore, the authors should give examples of specific applications, possibly to address a disease problem, to illustrate the future relevance of this research.

2 "Animal models integrate the host-microbe interactions but translation to the human situation remains limited due to major differences of digestive physiology, oral and gut microbiota between most animal models and human" Animal models can be constructed simultaneously with disease models. Is there any advantage to the authors' proposed approach to studying disease mechanisms.

3 Does the process of modeling oral to intestinal invasion simulate passing through the environment of gastric acid?

4 In this study, the authors only collected samples from two healthy donors. The results presented in this study are not statistically significant.

Reviewer #2 (Public repository details (Required)):

All sequencing data have been deposited at the European Nucleotide Archive database under the study accession PRJEB52431

Reviewer #2 (Comments for the Author):

This is a pilot study to assess the potential use of an in vitro model mucosal artificial colon (M-ARCOL), a one-stage fermentation system, to investigate the colonization of oral bacteria in the gut. Two fecal samples, one from each of the subjects, were individually used to inoculate the luminal compartment of the M-ARCOL system. Individual saliva samples were obtained from the same individuals to inoculate the mucosal compartment of the M-ARCOL system after the fecal samples were grown for eight days. Gas analysis and the analysis of the production of short-chain fatty acids were performed to characterize the system. Samples were obtained at different time points over 11 days for high-throughput sequencing. Metagenomic Species Pangenome (MSP) was used to identify and quantify microbial species. The results show the detection of oral taxa in the mucosal compartment of M-ARCOL. Overall, the experimental approach of the study is appropriate. The manuscript is well-written, and the results are interesting, supporting this model's use for future studies. There are a few comments to improve the manuscript.

Comments:

1. Fig 1. Why was it necessary to inoculate with the enriched saliva twice each day? Did the authors determine the relative amounts of retention of the taxa in the mucosal compartment after inoculation?
2. Line 206 and Table S2: Table S2 shows that 78 of 81 ND were not detected in either oral or stool/mucosal samples. Only three were detected in stool and saliva samples. Therefore, the statement "ND, because detected in stool and saliva samples" is not justified. It is possible that these taxa were not detected in the fecal samples but were detectable in the luminal compartment after cultivation. If so, the authors should clarify the statement.
3. Lines 215-222, Fig 4 and Fig S5: The results from Fig 4 and Fig 5S appear to be inconsistent. For example, Fig 5S shows the relative abundance and the presence of oral taxa in the luminal and mucosal compartments of donor S1 but not donor S2. However, Fig 4 shows MSP counts of oral taxa in the mucosal compartment of both donors. It is appreciated that the count and relative abundance may differ in their visibility in the bar charts. However, the authors should review these figures for possible errors.
4. Fig 5: it is unclear what is "D10_saliva." Is it the saliva before enrichment?
5. Presumably, the medium circulates between the luminal and mucosal compartments. The authors should explain why the oral taxa in the mucosal compartment were not detected in the luminal compartment.
6. A study's weakness is the lack of data regarding the amount of retention or the growth of the oral taxa in the system. The authors should provide some discussion of these issues.

Reviewer #4 (Comments for the Author):

Summary of review

The study by Etienne-Mesmin, L. and Meslier, V., et al. (Spectrum04344-22) described in the manuscript entitled, "In vitro modelling of oral microbial invasion in the human colon," details the authors' proof-of-principal study using a Mucosal Artificial Colon (M-ARCOL) system to investigate invasion of oral microbes into the gut. Invasion of oral bacteria into the gut has recently been identified to occur in disease states, and the contribution of oral bacterial colonization to such diseases in terms of contributing to pathologies or disease progression are poorly understood. Thus, a model to investigate such bacteria invasion and changes in microbiome composition will be useful to fill the current gap in knowledge. Here, the authors test the functionality

of the M-ARCOL system to assess oral-to-gut invasion. The authors start by collecting fecal samples from two donors that differ in age, diet, and oral bacterial composition (one methane-producing and one none) to use to inoculate respective M-ARCOL systems. After equilibration of the biomes in the luminal and mucosal compartments of the M-ARCOL system with the seeded fecal samples, the authors inoculate the system with bacteria enriched from saliva samples from the respective donors to mimic oral-to-gut invasion. The authors used next generation shotgun sequencing techniques to sequence isolated DNA to monitor and resolve species-level composition of the donor samples and the biome in the M-ARCOL system over time before and after simulated oral-gut-invasion. The authors observe that the oral bacteria mainly colonize the mucosal compartment of the M-ARCOL system, highlighting the functionality of the system by differentiating between the gut lumen and mucosal surfaces. Between the two donors, it was observed that several different oral species were able to colonize the mucosal compartment with the previously established fecal bacteria. In total, the authors propose that this proof-of-principal study supports the notion that the M-ARCOL system can be used as a system to assess in vitro oral-to-gut invasion to study oral bacterial invasion of the gut.

The manuscript is of sensible significance to the target scientific community, and describes an original idea of the authors' to utilize the M-ARCOL system as an in vitro model to assess invasion of oral bacteria into the gut. The manuscript explains a logical scientific approach and experimental design to address the hypothesis proposed by the authors. The experimental techniques and data analysis were clearly presented.

Overall enthusiasm for the manuscript is high, with the exception of several concerns. In particular, there is concern for the stated results and conclusion for Figures 4 and S5 given observed discrepancies between the data presented in the figures. Other concerns include the addition of comments to clarify limitations and caveats of this study and interpretations of the results.

Major points to address

-Figures 4 & S5: There appears to be significant discrepancies between Fig. 4 and Fig. S5 in terms of what the data show. As written and presented, it is unclear if these figures were generated from the same raw data (perhaps Supplemental Table 3?). It is also unclear why there are difference in the presence and absences in oral MSPs between the two figures. Specifically, Fig. 4 shows that in the S1 donor sample oral MSPs are absent in the luminal compartment throughout the experiment, while oral MSPs appear at D10, as expected. This is different from Fig S5 which shows in the S1 donor sample that oral MSPs are present in the luminal compartment at D7, a day before simulated oral invasion, then increase in the following days. There also appears to be a small abundance of oral MSPs in the S1 mucosal compartment on D8 shown in Fig. S5, which are unapparent in the Fig 4 S1 donor mucosal compartment D8 sample. It is a concern that in the S2 donor sample in Fig. 4 that oral MSPs are detected by D10, but it seems as though oral MSPs are absent from the mucosal compartment in Fig. 5. Additionally, lines 214-215 read, "... yet two oral MSP species were detected even before simulated oral-to-gut invasion (Fig. 4)," but there does not seem to be any visible (blue bar) oral MSPs present in either compartment from either donor before the addition of enriched saliva on D9. Please clarify which data were used to generate the figures, and address the concerns of the noted discrepancies in presented data. Perhaps revise the section in lines 201-222.

-In this study, the M-ARCOL system does not take into consideration the host aspect. Can the authors comment on this limitation and the potential adaptability of the M-ARCOL system to be seeded with intestinal cells or even highly differentiated intestinal organoids to better represent the host cells and secreted factors generated by these cells, and how these may affect the system? Host factors produced by intestinal epithelium may play a major role in interkingdom signaling, mucus and thus nutrient availability, and indicators of intestinal inflammation.

-Lines 360-362: This sentence indicates that beads were replaced every 2 days. Can the authors describe the M-ARCOL system in more detail with regard to the mucosal compartment? It is not clear why beads were replaced every two days. One would think that the bacteria in this compartment would be adhered to or growing on the beads, so wouldn't removing the beads and replacing the beads with new ones 'reset' the bacterial composition in this compartment? Or is it more so that the beads just provide a source of nutrients rather than nutrients and a substratum for bacteria? If the bacteria are not attaching to and growing on the beads, then does this system actually emulate the colon/mucosae, since there isn't a substratum (or more so a mucin-like gel matrix) to which the bacteria could adhere like they would in vivo, and are instead growing in suspension with the addition of Type-II mucin as a nutrient source? Please clarify.

-The in vitro fermentation reaction equilibrated after losing almost half the MSP by Day 8 (Fig. 2). The authors should explain the limitations and caveats associated with this observation when interpreting results and formulating their conclusions in the Discussion section. Please specifically comment on the fact that this loss in diversity may affect the invading salivary bacterial composition in the gut model (either by promoting or limiting oral bacterial colonization).

-Will the authors comment on their justification to stop the experiment just 2 days after the first enriched saliva introduction? It would be useful to see if the salivary microbes were able to persistently colonize the M-ARCOL system for at least as long as it took the stool sample bacteria to equilibrate (8 days) in the system.

-Lines 198-199: States, "No major changes in raw and enriched saliva samples were found within each donor, as confirmed by Bray-Curtis dissimilarity analysis (Fig. S3)." However, in Figure 4, the enriched-saliva samples show an increase in Eukaryota compared to the raw saliva. The authors should note this difference, and also comment as to why Eukaryota was detected in the

enriched-saliva samples from both donors?

-Lines 374-377: In the Experimental design and sampling section of the Materials and Methods, and in Fig. 1, the description of raw versus enriched saliva, volumes, and when enriched or raw saliva was injected is confusing. What is this last 5 ml of remaining late afternoon enriched saliva? Please clarify this explanation.

-On the website for the M-ARCOL system (<https://lyonbiopole.com/en/news/discover-mucosal-artificial-colon-m-arcol-obese-model>) the author Stéphanie Blanquet-Diot is cited as a director. Please clarify whether or not Lyonbiopole is a company, and please update the Competing Interests section accordingly to indicate Stéphanie Blanquet-Diot's affiliation with Lyonbiopole.

Additional suggested experiment

-It would be a worthwhile control to inoculate the enriched saliva samples in the M-ARCOL system alone to resolve any model-specific effects that are independent of the equilibrated stool microbiome.

Minor points to address

-Lines 371-372: The authors note that saliva samples contain host DNA, and that this is a problem for analyzing the microbial composition identified from sequences. Couldn't the authors instead filter-out reads that map to the human genome before processing the sequencing data? The authors did this with Bowtie2 so why was this such a big problem? The authors already performed a step to reduce the amount of host DNA in the sample, as noted in the methods. Though it is understood why the enrichment step was performed, enrichment adds yet another layer of in vitro manipulation to the workflow, and the effects of which may be amplified in the subsequent in vitro system. Though this is a proof-of-principle study, the authors should consider refining the protocol for the future, and discuss the caveats and limitations to the study presented herein be careful not to over interpret species-specific results.

-The nomenclature of "not-determined (ND)" seems to be a misnomer, because the niche is not undefined, rather there is more than one niche. It is recommended that instead of "ND" this group be labeled as "Both."

-Figure 2 & 3: Suggest separating the three samples by vertical (perhaps dashed) lines.

-Line 83: Should this be gut, instead of oral, samples? If it is in fact oral samples, why is there so much host DNA in oral samples compared to gut samples?

-Line 153: "(Fig. S3 C)." There is no panel C to this figure.

-Line 222 & Fig.4 : A comment should be added about the results pertaining to the enriched saliva of both donors, which appeared to show the presence of some 'ND' MSPs compared to the raw saliva.

-Figure 3: Please also mark on the graph when the saliva was added.

-Figure 3 & S4: It would be helpful for the reader if the legends were ordered so that the most abundant phyla in Fig., or families for S3, were written first. For example, in Fig. 3, Bacteroidota, Firmicutes, and Proteobacteria were listed first.

-Supplemental Figure S4: Can the authors increase the length of the Y-axis so it is easier to resolve some of the less-abundant families in the graph?

-Figure 4: The authors should include data for the ND samples in the first panel graphs in A and B representing the raw samples.

-Figure 5 and lines 225 to 228: On day 11 it appears that *Veillonella tobetsuensis* is also present in low abundance in Donor S1.

-Line 241: It would be helpful if the authors could provide a brief conclusion of the results identified by the analysis in the previous statement. For example, "These data suggest that abundance of oral invaders alone, but likely the species-specific phenotypes, permitted bacteria to invade the mucosa."

-Line 282: I think word 'confronting' should be 'emulating'

-Lines 389-391: Please state the rationale for using the two separate DNA isolation protocols for the fecal, saliva and luminal samples versus the mucosal samples.

-Line 398: The word 'shared' should be 'sheared.'

Reviewer #5 (Comments for the Author):

This is a technically sound study in the area of intestinal microbiology and demonstrates experimental and methodological rigour. However, it is important to attend to some recommendations:

1. RESULTS, line 109. A very methodological description is made, the result to be shown is not clear. it is important to leave the methodological details fully described in the methodology and not include them in the results as it is redundant.
2. The selection criteria of the fecal and saliva donors selected in this study are not clear. there is a significant age difference. how do you explain this
3. Although it is clarified that it is a non-interventionist study. The participation of humans as sample donors must require ethical supervision so that it can be classified as a study without risk, before an ethics committee, beyond assuming it.
4. There is no evidence of experimental controls including, for example, inoculation of the M-ARCOL system with sterile saliva. It is important to have population controls in the system to identify microorganisms in the samples and not outside.
5. Was a control carried out to verify that the inoculated bacteria are viable? sequencing identifies microorganisms in the sample regardless of their viability, which could greatly influence the expected results on the system.

Staff Comments:

Preparing Revision Guidelines

Please return the manuscript within 60 days; if you cannot complete the modification within this time period, please contact me. If you do not wish to modify the manuscript and prefer to submit it to another journal, please notify me of your decision immediately so that the manuscript may be formally withdrawn from consideration by Microbiology Spectrum.

The authors propose a new model of oral to gut invasion by the combined use of an in vitro model. This is a very interesting work. However, couples of concerns must be addressed by the authors before the considerable publication.

1 The authors claimed that their approach could provide useful mechanistic insights into the role of the oral microbiome in various disease processes. Therefore, the authors should give examples of specific applications, possibly to address a disease problem, to illustrate the future relevance of this research.

2 “Animal models integrate the host-microbe interactions but translation to the human situation remains limited due to major differences of digestive physiology, oral and gut microbiota between most animal models and human” Animal models can be constructed simultaneously with disease models. Is there any advantage to the authors' proposed approach to studying disease mechanisms.

3 Does the process of modeling oral to intestinal invasion simulate passing through the environment of gastric acid?

4 In this study, the authors only collected samples from two healthy donors. The results presented in this study are not statistically significant.

Summary of review

The study by Etienne-Mesmin, L. and Meslier, V., et al. (Spectrum04344-22) described in the manuscript entitled, “*In vitro* modelling of oral microbial invasion in the human colon,” details the authors’ proof-of-principal study using a Mucosal Artificial Colon (M-ARCOL) system to investigate invasion of oral microbes into the gut. Invasion of oral bacteria into the gut has recently been identified to occur in disease states, and the contribution of oral bacterial colonization to such diseases in terms of contributing to pathologies or disease progression are poorly understood. Thus, a model to investigate such bacteria invasion and changes in microbiome composition will be useful to fill the current gap in knowledge. Here, the authors test the functionality of the M-ARCOL system to assess oral-to-gut invasion. The authors start by collecting fecal samples from two donors that differ in age, diet, and oral bacterial composition (one methane-producing and one none) to use to inoculate respective M-ARCOL systems. After equilibration of the biomes in the luminal and mucosal compartments of the M-ARCOL system with the seeded fecal samples, the authors inoculate the system with bacteria enriched from saliva samples from the respective donors to mimic oral-to-gut evasion. The authors used next generation shotgun sequencing techniques to sequence isolated DNA to monitor and resolve species-level composition of the donor samples and the biome in the M-ARCOL system over time before and after simulated oral-gut-invasion. The authors observe that the oral bacteria mainly colonize the mucosal compartment of the M-ARCOL system, highlighting the functionality of the system by differentiating between the gut lumen and mucosal surfaces. Between the two donors, it was observed that several different oral species were able to colonize the mucosal compartment with the previously established fecal bacteria. In total, the authors propose that this proof-of-principal study supports the notion that the M-ARCOL system can be used as a system to assess *in vitro* oral-to-gut invasion to study oral bacterial invasion of the gut.

The manuscript is of sensible significance to the target scientific community, and describes an original idea of the authors’ to utilize the M-ARCOL system as an *in vitro* model to assess invasion of oral bacteria into the gut. The manuscript explains a logical scientific approach and experimental design to address the hypothesis proposed by the authors. The, experimental techniques and data analysis were clearly presented.

Overall enthusiasm for the manuscript is high, with the exception of several concerns. In particular, there is concern for the stated results and conclusion for Figures 4 and S5 given observed discrepancies between the data presented in the figures. Other concerns include the addition of comments to clarify limitations and caveats of this study and interpretations of the results.

Major points to address

-Figures 4 & S5: There appears to be significant discrepancies between Fig. 4 and Fig. S5 in terms of what the data show. As written and presented, it is unclear if these figures were generated from the same raw data (perhaps Supplemental Table 3?). It is also unclear why there are difference in the presence and absences in oral MSPs between the two figures. Specifically, Fig. 4 shows that in the S1 donor sample oral MSPs are absent in the luminal compartment throughout the experiment, while oral MSPs appear at D10, as expected. This is different from Fig S5 which shows in the S1 donor sample that oral MSPs are present in the luminal compartment at D7, a day before simulated oral invasion, then increase in the following days. There also appears to be a small abundance of oral MSPs in the S1 mucosal

compartment on D8 shown in Fig. S5, which are unapparent in the Fig 4 S1 donor mucosal compartment D8 sample. It is a concern that in the S2 donor sample in Fig. 4 that oral MSPs are detected by D10, but it seems as though oral MSPs are absent from the mucosal compartment in Fig. 5. Additionally, lines 214-215 read, "... yet two oral MSP species were detected even before simulated oral-to-gut invasion (Fig. 4)," but there does not seem to be any visible (blue bar) oral MSPs present in either compartment from either donor before the addition of enriched saliva on D9. Please clarify which data were used to generate the figures, and address the concerns of the noted discrepancies in presented data. Perhaps revise the section in lines 201-222.

-In this study, the M-ARCOL system does not take into consideration the host aspect. Can the authors comment on this limitation and the potential adaptability of the M-ARCOL system to be seeded with intestinal cells or even highly differentiated intestinal organoids to better represent the host cells and secreted factors generated by these cells, and how these may affect the system? Host factors produced by intestinal epithelium may play a major role in interkingdom signaling, mucus and thus nutrient availability, and indicators of intestinal inflammation.

-Lines 360-362: This sentence indicates that beads were replaced every 2 days. Can the authors describe the M-ARCOL system in more detail with regard to the mucosal compartment? It is not clear why beads were replaced every two days. One would think that the bacteria in this compartment would be adhered to or growing on the beads, so wouldn't removing the beads and replacing the beads with new ones 'reset' the bacterial composition in this compartment? Or is it more so that the beads just provide a source of nutrients rather than nutrients and a substratum for bacteria? If the bacteria are not attaching to and growing on the beads, then does this system actually emulate the colon/mucosae, since there isn't a substratum (or more so a mucin-like gel matrix) to which the bacteria could adhere like they would *in vivo*, and are instead growing in suspension with the addition of Type-II mucin as a nutrient source? Please clarify.

-The *in vitro* fermentation reaction equilibrated after losing almost half the MSP by Day 8 (Fig. 2). The authors should explain the limitations and caveats associated with this observation when interpreting results and formulating their conclusions in the Discussion section. Please specifically comment on the fact that this loss in diversity may affect the invading salivary bacterial composition in the gut model (either by promoting or limiting oral bacterial colonization).

-Will the authors comment on their justification to stop the experiment just 2 days after the first enriched saliva introduction? It would be useful to see if the salivary microbes were able to persistently colonize the M-ARCOL system for at least as long as it took the stool sample bacteria to equilibrate (8 days) in the system.

-Lines 198-199: States, "No major changes in raw and enriched saliva samples were found within each donor, as confirmed by Bray-Curtis dissimilarity analysis (Fig. S3)." However, in Figure 4, the enriched-saliva samples show an increase in Eukaryota compared to the raw saliva. The authors should note this difference, and also comment as to why Eukaryota was detected in the enriched-saliva samples from both donors?

-Lines 374-377: In the Experimental design and sampling section of the Materials and Methods, and in Fig. 1, the description of raw versus enriched saliva, volumes, and when enriched or raw saliva was injected is confusing. What is this last 5 ml of remaining late afternoon enriched saliva? Please clarify this explanation.

-On the website for the M-ARCOL system (<https://lyonbiopole.com/en/news/discover-mucosal-artificial-colon-m-arcol-obese-model>) the author Stéphanie Blanquet-Diot is cited as a director. Please clarify whether or not Lyonbiopole is a company, and please update the Competing Interests section accordingly to indicate Stéphanie Blanquet-Diot's affiliation with Lyonbiopole.

Additional suggested experiment

-It would be a worthwhile control to inoculate the enriched saliva samples in the M-ARCOL system alone to resolve any model-specific effects that are independent of the equilibrated stool microbiome.

Minor points to address

-Lines 371-372: The authors note that saliva samples contain host DNA, and that this is a problem for analyzing the microbial composition identified from sequences. Couldn't the authors instead filter-out reads that map to the human genome before processing the sequencing data? The authors did this with Bowtie2 so why was this such a big problem? The authors already performed a step to reduce the amount of host DNA in the sample, as noted in the methods. Though it is understood why the enrichment step was performed, enrichment adds yet another layer of *in vitro* manipulation to the workflow, and the effects of which may be amplified in the subsequent *in vitro* system. Though this is a proof-of-principle study, the authors should consider refining the protocol for the future, and discuss the caveats and limitations to the study presented herein be careful not to over interpret species-specific results.

-The nomenclature of "not-determined (ND)" seems to be a misnomer, because the niche is not undefined, rather there is more than one niche. It is recommended that instead of "ND" this group be labeled as "Both."

-Figure 2 & 3: Suggest separating the three samples by vertical (perhaps dashed) lines.

-Line 83: Should this be gut, instead of oral, samples? If it is in fact oral samples, why is there so much host DNA in oral samples compared to gut samples?

-Line 153: "(Fig. S3 C)." There is no panel C to this figure.

-Line 222 & Fig.4 : A comment should be added about the results pertaining to the enriched saliva of both donors, which appeared to show the presence of some 'ND' MSPs compared to the raw saliva.

-Figure 3: Please also mark on the graph when the saliva was added.

-Figure 3 & S4: It would be helpful for the reader if the legends were ordered so that the most abundant phyla in Fig., or families for S3, were written first. For example, in Fig. 3, Bacteroidota, Firmicutes, and Proteobacteria were listed first.

-Supplemental Figure S4: Can the authors increase the length of the Y-axis so it is easier to resolve some of the less-abundant families in the graph?

-Figure 4: The authors should include data for the ND samples in the first panel graphs in A and B representing the raw samples.

-Figure 5 and lines 225 to 228: On day 11 it appears that *Veillonella tobetsuensis* is also present in low abundance in Donor S1.

-Line 241: It would be helpful if the authors could provide a brief conclusion of the results identified by the analysis in the previous statement. For example, "These data suggest that abundance of oral invaders alone, but likely the species-specific phenotypes, permitted bacteria to invade the mucosa."

-Line 282: I think word 'confronting' should be 'emulating'

-Lines 389-391: Please state the rationale for using the two separate DNA isolation protocols for the fecal, saliva and luminal samples versus the mucosal samples.

-Line 398: The word 'shared' should be 'sheared.'

Re: Spectrum04344-22 (In vitro modelling of oral microbial invasion in the human colon)

Dear Dr. Mathieu Almeida:

Thank you for submitting your manuscript to Microbiology Spectrum.

Sincerely,

Diyang Li

Journals Department
Point by point to reviewers' comments

We thank the reviewers for their careful and thoughtful analysis of our manuscript and are pleased to have addressed their specific concerns as an opportunity to improve our manuscript.

Reviewer #1:

The authors propose a new model of oral to gut invasion by the combined use of an *in vitro* model. This is a very interesting work. However, couples of concerns must be addressed by the authors before the considerable publication.

We thank reviewer #1 for his/her time and effort in reviewing our manuscript. We have tried to answer his/her concerns to the best of our abilities and hope it will meet his/her expectations.

1. The authors claimed that their approach could provide useful mechanistic insights into the role of the oral microbiome in various disease processes. Therefore, the authors should give examples of specific applications, possibly to address a disease problem, to illustrate the future relevance of this research.

We thank the reviewer for pointing this crucial point to our attention. It is important to note that the oral–gut microbiome axis is implicated in the pathogenesis of intestinal diseases such as inflammatory bowel diseases (e.g. Crohn disease (Hu et al. 2021)) or liver diseases (e.g. NAFLD, NASH, cirrhosis (Albuquerque-Souza and Sahingur 2022)), extra-intestinal disorders (e.g. obesity (Gasmi Benahmed et al. 2021)) but also in cancer (e.g. colorectal cancer, pancreatic carcinoma, (Park et al. 2021)). Several independent studies have shown that oral microbiota dysbiosis aggravates these pathologies possibly through modulation of the gut microbiota.

In this context, colon *in vitro* gut models, including M-ARCOL, can be adapted to mimic pathological situations associated to gut microbial dysbiosis, such as obesity, irritable bowel syndrome or IBD (as mentioned in the discussion **lines 335-337**). This can be performed by inoculating these models with fecal samples from patients but also by adapting all the nutritional and/or physicochemical parameters to the diseased situations. Understanding the correlation of the oral–gut microbiome axis in pathogenesis confers an advantage for precise diagnosis and effective treatment for example *via* targeted microbial strategies such as probiotics or fecal microbiota transplantation. We have added in the revised version of the discussion, the point raised by the reviewer (**lines 340-346**).

2. "Animal models integrate the host-microbe interactions but translation to the human situation remains limited due to major differences of digestive physiology, oral and gut microbiota between most animal models and human" Animal models can be constructed simultaneously with disease models. Is there any advantage to the authors' proposed approach to studying disease mechanisms.

We thank the reviewer for his/her comment. *In vitro* models integrating gut microbiome are usually inoculated with fecal samples from healthy subjects and set-up to reproduce digestive colonic conditions from healthy adult individuals. One of the main advantages of such models is the possibility to inoculate the bioreactors with fecal samples from diseased-patients and to modulate the nutritional and/or physicochemical parameters (e.g. pH, residence time) of the models to the specific colonic environment of a disease situation, since such factors are well known to shape gut microbiota composition (Pham et al. 2019). This information is mentioned **lines 335-337** of the discussion section. Another

advantage that we mentioned in our manuscript **lines 107-110** is that *in vitro* models are very good alternatives to animal models, that currently meet the 3R principles and EU regulations on animal research (https://ec.europa.eu/environment/chemicals/lab_animals/legislation_en.htm).

3. Does the process of modeling oral to intestinal invasion simulate passing through the environment of gastric acid?

We agree with the reviewer that a complete model of the upper GIT would be of interest. Meanwhile, the purpose of the present study was to investigate how oral microorganisms can invade the colonic compartment using the M-ARCOL model. To date, this system does not integrate the stomach and small intestinal compartment. As mentioned **lines 327-329** of our manuscript, future developments could include the implementation of an upper *in vitro* human digestive tract by coupling the M-ARCOL with the TIM-1 stomach and small intestinal digester.

4. In this study, the authors only collected samples from two healthy donors. The results presented in this study are not statistically significant.

We thank the reviewer for his/her comment and we agree that this study is preliminary but provided original evidence to support the use of the *in vitro* gut model as a valuable platform to study oral to gut invasion. Our study relies on using the microbiota from two healthy donors that differ in terms of gender (male *versus* female), age (22 *versus* 52 years old), methane status (one producer *versus* one non-producer) and diet (one consuming a western like diet *versus* one eating a flexitarian-based diet).

To go further, we performed a PCoA analysis (see figure below) based on Bray-Curtis distance matrix between our donors and French healthy individuals of the MetaCardis Cohort (Fromentin et al. 2022), and the results show that the donors selected in our study are representatives of two distinct clusters when considering fiber intake (dark green = high fiber intake > 23 g/day, red = low fiber intake ≤ 23g/day).

We think that the results of the present study, which is the first study of oral to gut microbial invasion using M-ARCOL model, represents an important data set that can be used to base further experiments using multiple donors to validate our findings.

Reviewer #2

This is a pilot study to assess the potential use of an in vitro model mucosal artificial colon (M-ARCOL), a one-stage fermentation system, to investigate the colonization of oral bacteria in the gut. Two fecal samples, one from each of the subjects, were individually used to inoculate the luminal compartment of the M-ARCOL system. Individual saliva samples were obtained from the same individuals to inoculate the mucosal compartment of the M-ARCOL system after the fecal samples were grown for eight days. Gas analysis and the analysis of the production of short-chain fatty acids were performed to characterize the system. Samples were obtained at different time points over 11 days for high-throughput sequencing. Metagenomic Species Pangenome (MSP) was used to identify and quantify microbial species. The results show the detection of oral taxa in the mucosal compartment of M-ARCOL. Overall, the experimental approach of the study is appropriate. The manuscript is well-written, and the results are interesting, supporting this model's use for future studies. There are a few comments to improve the manuscript.

We thank reviewer #2 for his/her enriching feedback and we are delighted to answer his/her concerns/questions.

1. Fig 1. Why was it necessary to inoculate with the enriched saliva twice each day?

In humans, the normal daily production of saliva varies between 0.5 and 1.5 liters (Iorgulescu 2009) which continuously flows in the gut. For evident technical reason, inoculating such a large amount of saliva was not possible, we made a compromise with two inoculations of enriched saliva in our study.

Did the authors determine the relative amounts of retention of the taxa in the mucosal compartment after inoculation?

In our prototype, we stopped the saliva injection after collecting samples at D10. We observed that some oral species in the luminal compartment were detected at D11 (Figure 5), even in the absence of oral injection. This observation suggests that future experiments should be extended for at least 2 days after the final saliva injection to measure the retention of oral species in the mucosal compartment in absence of saliva injection. We implemented this aspect in the manuscript **lines 331-333**.

2. Line 206 and Table S2: Table S2 shows that 78 of 81 ND were not detected in either oral or stool/mucosal samples. Only three were detected in stool and saliva samples. Therefore, the statement "ND, because detected in stool and saliva samples" is not justified. It is possible that these taxa were not detected in the fecal samples but were detectable in the luminal compartment after cultivation. If so, the authors should clarify the statement.

We thank the reviewer for pointing this out. We corrected the statement as follow: "ND, species that are either undetected in raw stool and raw saliva samples or detected in both stool and saliva samples before inoculation of the bioreactor at initial time point (T0). We modified this statement **lines 224-226** and in the method section **lines 472-474**.

3. Lines 215-222, Fig 4 and Fig S5: The results from Fig 4 and Fig 5S appear to be inconsistent. For example, Fig 5S shows the relative abundance and the presence of oral taxa in the luminal and mucosal compartments of donor S1 but not donor S2. However, Fig 4 shows MSP counts of oral taxa in the mucosal compartment of both donors. It is appreciated that the count and relative abundance may differ in their visibility in the bar charts. However, the authors should review these figures for possible errors.

We thank the reviewer for raising this observation. We want to clarify the difference between Figure 4 and Supplementary Figure 5. Figure 4 is related to the MSP counts, i.e. the number of detected MSP species for each ecological niche, while the Figure S5 is the relative abundance of each ecological niche, containing various number of MSP.

This discrepancy is related to the representation using the ggpubr package and the biostatistical R tool, when rendering small numbers (here, minimum MSP species = 1 for Figure 4) over large numbers (here, maximum MSP species = 326 for the entire dataset). Modifying the width and height solved this mis-representation and we provided a new version of Figure 4 in the revised version of the manuscript (see also revised Figure 4 below). We also provided a Supplementary Table 3 (new Table S3) that gives a complete overview of the data and we modified the manuscript **lines 231-234** accordingly.

Revised Figure 4 modifying height & width (this figure has been included in the revised manuscript):

4. Fig 5: it is unclear what is "D10_saliva." Is it the saliva before enrichment?

We apologize for this misunderstanding. Indeed, "D10_saliva" is a raw saliva. This specific sample was not enriched but processed in parallel of the D10 enriched saliva sample, as a comparative. We compared the composition of the saliva and enriched saliva samples from both donors using Bray-Curtis dissimilarity distances (plot below and Supplementary Figure S3) and we showed that the microbial structure of oral microbiome was conserved.

5. Presumably, the medium circulates between the luminal and mucosal compartments. The authors should explain why the oral taxa in the mucosal compartment were not detected in the luminal compartment.

We thank the reviewer for pointing this out. A study from Van der Hoeven and colleagues showed that oral species (especially *Streptococcus*) were capable to grow in vitro in defined medium containing pig gastric mucin and used this as a nutritional substrate. Studies also reported that patients with diseases of the gut (namely Crohn's disease or colorectal cancer) exhibit an abnormal enrichment of typical oral bacteria in the gut mucosal tissues (Gevers et al. 2014; Yachida et al. 2019). In our study, as mentioned **lines 295-298**, we can hypothesize that the oral taxa probably maintain better in the mucosal compartment since they are more adapted to such microenvironment.

6. A study's weakness is the lack of data regarding the amount of retention or the growth of the oral taxa in the system. The authors should provide some discussion of these issues.

We agree with the reviewer comment. It would be needed to investigate the maintenance of oral taxa within the M-ARCOL system by extending the experiment for at least 2 days after the last saliva injection. We have added in the revised version of the discussion the point raised by the reviewer **lines 331-333**.

Reviewer #4

The study by Etienne-Mesmin, L. and Meslier, V., et al. (Spectrum04344-22) described in the manuscript entitled, "In vitro modelling of oral microbial invasion in the human colon," details the authors' proof-of-principal study using a Mucosal Artificial Colon (M-ARCOL) system to investigate invasion of oral microbes into the gut. Invasion of oral bacteria into the gut has recently been identified to occur in disease states, and the contribution of oral bacterial colonization to such diseases in terms of contributing to pathologies or disease progression are poorly understood. Thus, a model to investigate such bacteria invasion and changes in microbiome composition will be useful to fill the current gap in knowledge. Here, the authors test the functionality of the M-ARCOL system to assess oral-to-gut invasion. The authors start by collecting fecal samples from two donors that differ in age, diet, and oral bacterial composition (one methane-producing and one none) to use to inoculate respective M-ARCOL systems. After equilibration of the biomes in the luminal and mucosal compartments of the M-ARCOL system with the seeded fecal samples, the authors inoculate the system with bacteria enriched from saliva samples from the respective donors to mimic oral-to-gut evasion. The authors used next generation shotgun sequencing techniques to sequence isolated DNA to monitor and resolve species-level composition of the donor samples and the

biome in the M-ARCOL system over time before and after simulated oral-gut-invasion. The authors observe that the oral bacteria mainly colonize the mucosal compartment of the M-ARCOL system, highlighting the functionality of the system by differentiating between the gut lumen and mucosal surfaces. Between the two donors, it was observed that several different oral species were able to colonize the mucosal compartment with the previously established fecal bacteria. In total, the authors propose that this proof-of-principal study supports the notion that the M-ARCOL system can be used as a system to assess in vitro oral-to-gut invasion to study oral bacterial invasion of the gut.

The manuscript is of sensible significance to the target scientific community, and describes an original idea of the authors' to utilize the M-ARCOL system as an in vitro model to assess invasion of oral bacteria into the gut. The manuscript explains a logical scientific approach and experimental design to address the hypothesis proposed by the authors. The experimental techniques and data analysis were clearly presented. Overall enthusiasm for the manuscript is high, with the exception of several concerns. In particular, there is concern for the stated results and conclusion for Figures 4 and S5 given observed discrepancies between the data presented in the figures. Other concerns include the addition of comments to clarify limitations and caveats of this study and interpretations of the results.

We thank the reviewer for his/her overall positive tone regarding our manuscript and for the interesting remarks and suggestions that will allow us to improve the quality of our manuscript.

Major points to address

-Figures 4 & S5: There appears to be significant discrepancies between Fig. 4 and Fig. S5 in terms of what the data show. As written and presented, it is unclear if these figures were generated from the same raw data (perhaps Supplemental Table 3?).

We apologize for this misunderstanding and thank the reviewer for pointing out that our Supplementary Table S3 was not mentioned in the manuscript. We have modified our manuscript accordingly **lines 224-229** to refer to this table when appropriate.

It is also unclear why there are difference in the presence and absences in oral MSPs between the two figures. Specifically, Fig. 4 shows that in the S1 donor sample oral MSPs are absent in the luminal compartment throughout the experiment, while oral MSPs appear at D10, as expected. This is different from Fig S5 which shows in the S1 donor sample that oral MSPs are present in the luminal compartment at D7, a day before simulated oral invasion, then increase in the following days. There also appears to be a small abundance of oral MSPs in the S1 mucosal compartment on D8 shown in Fig. S5, which are unapparent in the Fig 4 S1 donor mucosal compartment D8 sample. It is a concern that in the S2 donor sample in Fig. 4 that oral MSPs are detected by D10, but it seems as though oral MSPs are absent from the mucosal compartment in Fig. 5. Additionally, lines 214-215, "... yet two oral MSP species were detected even before simulated oral-to-gut invasion (Fig. 4)," but there does not seem to be any visible (blue bar) oral MSPs present in either compartment from either donor before the addition of enriched saliva on D9. Please clarify which data were used to generate the figures, and address the concerns of the noted discrepancies in presented data. Perhaps revise the section in lines 201-222.

We totally agree with the reviewer for this observation. This discrepancy is related to the representation using the ggpubr package and the biostatistical R tool, when rendering small numbers (here, minimum MSP species = 1 for Figure 4) over large numbers (here, maximum MSP species = 326 for the entire dataset). Both figures were done from the same data, that we reported in the revised version of our manuscript with the Supplementary Table S3 for clarification. We also modified the width and height to solve this mis-representation and we provided a new version of Figure 4 in the revised version of the manuscript (see also revised Figure 4 below).

Revised Figure 4 modifying height & width (this figure has been included in the revised manuscript):

-In this study, the M-ARCOL system does not take into consideration the host aspect. Can the authors comment on this limitation and the potential adaptability of the M-ARCOL system to be seeded with intestinal cells or even highly differentiated intestinal organoids to better represent the host cells and secreted factors generated by these cells, and how these may affect the system? Host factors produced by intestinal epithelium may play a major role in interkingdom signaling, mucus and thus nutrient availability, and indicators of intestinal inflammation.

We thank the reviewer for his/her comment and we agree that this model has some limitations, such as the lack of host interactions (e.g. no immune, hormonal or nervous systems) and feedback mechanisms. In order to get closer to the *in vivo* situation by integrating host-microbiota interactions, current technological challenges aim to couple *in vitro* colon models to intestinal epithelial cells or immune cells. *In vitro* gut models can also be coupled to more complex units, such as intestinal organoids (Kim et al. 2020) or bioengineered human gut-on-chip devices such as HuMiX (Shah et al. 2016), Intestine-Chip (Kim et al. 2016) and Colon-Chip systems. We have added in the revised version of the discussion this raised by the reviewer **lines 340-346**.

-Lines 360-362: This sentence indicates that beads were replaced every 2 days. Can the authors describe the M-ARCOL system in more detail with regard to the mucosal compartment? It is not clear why beads were replaced every two days. One would think that the bacteria in this compartment would be adhered to or growing on the beads, so wouldn't

removing the beads and replacing the beads with new ones 'reset' the bacterial composition in this compartment? Or is it more so that the beads just provide a source of nutrients rather than nutrients and a substratum for bacteria? If the bacteria are not attaching to and growing on the beads, then does this system actually emulate the colon/mucosae, since there isn't a substratum (or more so a mucin-like gel matrix) to which the bacteria could adhere like they would in vivo, and are instead growing in suspension with the addition of Type-II mucin as a nutrient source? Please clarify.

All the information regarding mucin-alginate beads are already mentioned in the methods section of our manuscript (lines 390-400).

To go further, mucin-alginate beads were totally replaced every 48h to avoid beads degradation by gut microbes and to ensure a continuous availability of mucin adherent surfaces. On a technical point of view, medium surrounding the mucin/alginate beads is collected, then mucin/alginate beads are replaced every 48 hours, upon collection they were gently washed 3 times in sterile PBS 1X so only bacteria able to adhere to mucin/alginate beads are recovered. Fresh mucin/alginate beads are added to the external vessel and the surrounding medium is reinjected into the vessel to avoid the reset of bacterial composition in the mucosal compartment.

-The *in vitro* fermentation reaction equilibrated after losing almost half the MSP by Day 8 (Fig. 2). The authors should explain the limitations and caveats associated with this observation when interpreting results and formulating their conclusions in the Discussion section. Please specifically comment on the fact that this loss in diversity may affect the invading salivary bacterial composition in the gut model (either by promoting or limiting oral bacterial colonization).

Inoculation of *in vitro* gut models with fecal samples aims to reproduce the complexity and diversity of the microbial ecosystem encountered within the human colon. M-ARCOL model enables to conserve the microbial composition of initial fecal sample at the individual scale (see plot below).

Nonetheless, whatever the *in vitro* gut models considered, a decrease of bacterial diversity compared to initial fecal sample is observed in the models (for M-ARCOL, (Deschamps et al. 2020), for M-SHIME, (Van den Abbeele et al. 2013); for PolyFermS, (Zihler Berner et al. 2013)). Simulating a mucosal environment by the addition of mucin-alginate beads avoids the wash-out of surface-attached bacteria, indeed sustaining the maintenance at high levels of bacteria in this specific compartment as shown for M-SHIME

model (Van den Abbeele et al. 2013) and M-ARCOL (Deschamps et al. 2020; Fournier et al. 2023).

In the context of various intestinal disorders (IBD, IBS) or extra-intestinal disorders (obesity, liver diseases), loss of gut bacterial diversity has been widely documented in the literature (Mosca et al. 2016) and associated with the invasion of oral microorganisms. In our study, we showed that although a loss of richness was observed in the luminal compartment, the invasion of oral species occurred mostly in the mucosal compartment, that is always richer than the luminal compartment (for equivalent time point). These results have never been reported before and suggest that the mucin preference is equally or more important than the microbial richness in the invasion process.

-Will the authors comment on their justification to stop the experiment just 2 days after the first enriched saliva introduction? It would be useful to see if the salivary microbes were able to persistently colonize the M-ARCOL system for at least as long as it took the stool sample bacteria to equilibrate (8 days) in the system.

We agree with the reviewer comment. It would be of importance to study the invasion of oral microbes in the *in vitro* M-ARCOL model for a longer period. We have added in the revised version of the discussion the point raised by the reviewer **lines 331-333**.

-Lines 198-199: States, "No major changes in raw and enriched saliva samples were found within each donor, as confirmed by Bray-Curtis dissimilarity analysis (Fig. S3)." However, in Figure 4, the enriched-saliva samples show an increase in Eukaryota compared to the raw saliva. The authors should note this difference, and also comment as to why Eukaryota was detected in the enriched-saliva samples from both donors?

We thank the reviewer for pointing this out. This was also a counter-intuitive and intriguing observation for us. Potentially, human DNA remained during saliva enrichment by the formation of microbial aggregates seeding upon human salivary mucosa and associated host cells, as observed recently by Simon-Soro and colleagues (Simon-Soro et al. 2022). We added this hypothesis in the discussion **lines 295-298**.

-Lines 374-377: In the Experimental design and sampling section of the Materials and Methods, and in Fig. 1, the description of raw versus enriched saliva, volumes, and when enriched or raw saliva was injected is confusing. What is this last 5 ml of remaining late afternoon enriched saliva? Please clarify this explanation.

We thank the reviewer for raising this question. 10 mL of saliva was collected twice a day (morning and afternoon) in a sterile container. For saliva enrichment, 10mL of freshly collected saliva was resuspended in 15 mL of colonic nutritive medium to favor oral bacteria multiplication and enrich the microbial fraction over the human fraction in the saliva samples. As a result, the final volume was 25 mL after amplification and 10 mL of cultured saliva was injected twice a day (the morning saliva sample was enriched for 9 hours and then injected in the evening of the same day in M-ARCOL (10ml), the evening saliva sample was enriched for 9 hours and then was injected the following morning in M-ARCOL (10 ml)). The 5 ml of remaining were used to characterize enriched saliva, this has been updated in the revised version of the manuscript (**line 412-415**).

-On the website for the M-ARCOL system (<https://lyonbiopole.com/en/news/discover-mucosal-artificial-colon-m-arcobese-model>) the author Stéphanie Blanquet-Diot is cited as a director. Please clarify whether or not Lyonbiopole is a company, and please update the Competing Interests section accordingly to indicate Stéphanie Blanquet-Diot's affiliation with Lyonbiopole.

To make it clearer, Lyonbiopole is a competitive cluster that advises and works with more than 270 members, companies, academics and hospitals in the Auvergne-Rhône-Alpes region, for their innovation, growth or hosting projects. Dr. Stephanie Blanquet Diot is an active member of the Cluster, and she is the deputy director of her research unit (UMR MEDIS, Université Clermont Auvergne, FRANCE).

Additional suggested experiment

-It would be a worthwhile control to inoculate the enriched saliva samples in the M-ARCOL system alone to resolve any model-specific effects that are independent of the equilibrated stool microbiome.

We thank the reviewer for this suggestion, but M-ARCOL model is not adapted to the specific physicochemical conditions of the oral environment in terms of pH, nutritive medium, residence time, anaerobiosis. Indeed, the M-ARCOL reproduces the specific physicochemical and microbial conditions encountered in the human colonic environment, we do not think that the addition of enriched saliva itself would lead to the identification of model-specific effects as it is not adapted to the oral conditions.

Minor points to address

-Lines 371-372: The authors note that saliva samples contain host DNA, and that this is a problem for analyzing the microbial composition identified from sequences. Couldn't the authors instead filter-out reads that map to the human genome before processing the sequencing data? The authors did this with Bowtie2 so why was this such a big problem? The authors already performed a step to reduce the amount of host DNA in the sample, as noted in the methods. Though it is understood why the enrichment step was performed, enrichment adds yet another layer of in vitro manipulation to the workflow, and the effects of which may be amplified in the subsequent in vitro system. Though this is a proof-of-principle study, the authors should consider refining the protocol for the future, and discuss the caveats and limitations to the study presented herein be careful not to over interpret species-specific results.

We thank the reviewer for this comment. The suggested bowtie2 filtering was indeed performed as noticed by the reviewer to remove human related sequences. Here, we pointed out that classical shotgun metagenomic approaches require to sequence all DNA and then filter out human DNA, which in saliva represent up to 90% of the sequenced DNA. One way to solve this issue would be to improve saliva sample treatment to reduce human DNA presence before sequencing, as proposed by Marotz et al (Marotz et al. 2018). However, this could induce biases in the interpretation as aggregated prokaryotic cells seeding upon host cells could be removed during the filtering step. This was also observed by Marotz et al, showing that all existing filtering methods (Fil, NEB, Mol, QIA, lyPMA) increase microbial compositional dissimilarity compared to raw sample. We modified the manuscript accordingly **lines 295-298**.

-The nomenclature of "not-determined (ND)" seems to be a misnomer, because the niche is not undefined, rather there is more than one niche. It is recommended that instead of "ND" this group be labeled as "Both."

We thank the reviewer for pointing this misunderstanding, and acknowledge that one part of the "ND" definition was missing in the manuscript. We completed the definition in the manuscript **lines 224-226** and **lines 472-474** as follow: "ND, species that are either undetected in raw stool and raw saliva samples or detected in both stool and saliva samples before inoculation of the bioreactor at the initial time point (T0). We also refer to the Supplementary Table S2, providing the complete ecological niche definition.

-Figure 2 & 3: Suggest separating the three samples by vertical (perhaps dashed) lines.

We modified the figures accordingly.

-Line 83: Should this be gut, instead of oral, samples? If it is in fact oral samples, why is there so much host DNA in oral samples compared to gut samples?

We confirm to the reviewer that this section concerns oral microbial communities. Indeed, as mentioned by Marotz et al: "The proportion of human cells to microbial cells varies widely by sampling site" (Marotz et al. 2018). For example, fecal samples from healthy controls typically yield < 10% human genome-aligned reads, but human saliva, nasal cavity, skin, and vaginal samples routinely contain >90%." This over representation of human-related reads is indeed a characteristic of oral samples in our context of studying healthy individuals' samples.

-Line 153: "(Fig. S3 C)." There is no panel C to this figure.

We apologize for this discrepancy; we indeed refer to the Figure S3. This mistake has been corrected (**line 169**) of the revised manuscript.

-Line 222 & Fig.4 : A comment should be added about the results pertaining to the enriched saliva of both donors, which appeared to show the presence of some 'ND' MSPs compared to the raw saliva.

While our primary presentation of the results was to focus on the luminal and mucosal compartments, we acknowledge this comment. While we have updated the definition of ND species **lines 224-226**, we first clarified that some ND species were detected in the raw stool and saliva samples **lineS 2296230**, and modified our manuscript to acknowledge the detection of ND species in the enriched saliva samples **lines 231-234**.

-Figure 3: Please also mark on the graph when the saliva was added.

We added arrows for saliva injection in the revised Figure 3

-Figure 3 & S4: It would be helpful for the reader if the legends were ordered so that the most abundant phyla in Fig., or families for S3, were written first. For example, in Fig. 3, Bacteroidota, Firmicutes, and Proteobacteria were listed first.

-Supplemental Figure S4: Can the authors increase the length of the Y-axis so it is easier to resolve some of the less-abundant families in the graph?

According to reviewer's suggestion, we have revised the Figure 3 and Figure S4 accordingly, ranking each taxon by means.

-Figure 4: The authors should include data for the ND samples in the first panel graphs in A and B representing the raw samples.

ND species are represented in the revised Figure 4.

-Figure 5 and lines 225 to 228: On day 11 it appears that Veillonella tobetsuensis is also present in low abundance in Donor S1.

We acknowledged this comment **lines 262-264** of the revised manuscript.

-Line 241: It would be helpful if the authors could provide a brief conclusion of the results identified by the analysis in the previous statement. For example, "These data suggest that

abundance of oral invaders alone, but likely the species-specific phenotypes, permitted bacteria to invade the mucosa."

We thank the reviewer for this comment and add this sentence **lines 313-314**.

-Line 282: I think word 'confronting' should be 'emulating'

We thank the reviewer for this comment and we have changed accordingly.

-Lines **389-391**: Please state the rationale for using the two separate DNA isolation protocols for the fecal, saliva and luminal samples versus the mucosal samples.

We thank the reviewer for this suggestion. Indeed, we used a protocol for the mucosal samples that is more adapted for high microbial biomass samples, using column from the QiaAmp protocol from Qiagen, while the protocol used for the other samples is more adapted for low microbial biomass and recover DNA by precipitation. We added this information **lines 421 and 430-431** of the revised manuscript.

-Line 398: The word 'shared' should be 'sheared.'

We have changed accordingly.

Reviewer #5

This is a technically sound study in the area of intestinal microbiology and demonstrates experimental and methodological rigour. However, it is important to attend to some recommendations:

1. RESULTS, line 109. A very methodological description is made, the result to be shown is not clear. It is important to leave the methodological details fully described in the methodology and not include them in the results as it is redundant.

We thank the reviewer for his/her comment. To our knowledge, our work provided original evidence to support the use of the *in vitro* gut model as a valuable platform to study oral to gut invasion and it appears important to clearly explain for the readers the set-up of the model both in the methods and discussion sections.

2. The selection criteria of the fecal and saliva donors selected in this study are not clear. There is a significant age difference. How do you explain this?

We thank the reviewer for his/her comment and we agree that this study is preliminary but provided original evidence to support the use of the *in vitro* gut model as a valuable platform to study oral to gut invasion. Our study relies on using the microbiota from two healthy donors that differ in terms of gender (male *versus* female), age (22 *versus* 52 years old), methane status (one producer *versus* one non-producer) and diet (one consuming a western like diet *versus* one eating a flexitarian-based diet).

To go further, we performed a PCoA analysis (see figure below) based on Bray-Curtis distance matrix between our donors and French healthy individuals of the MetaCardis Cohort (Fromentin et al. 2022), and the results show that the donors selected in our study are representatives of two distinct clusters when considering fiber intake (dark green = high fiber intake > 23 g/day, red = low fiber intake ≤ 23g/day).

We think that the results of the present study, which is the first study of oral to gut microbial invasion using M-ARCOL model, represents an important data set that can be used to base further experiments using multiple donors to validate our findings.

3. Although it is clarified that it is a non-interventionist study. The participation of humans as sample donors must require ethical supervision so that it can be classified as a study without risk, before an ethics committee, beyond assuming it.

This study is a non-interventional study with no additions to usual clinical cares, as reported in the ethics declaration. According to the French Public Health Code, we obtained specific written informed consent from the donors and the protocol did not require approval from an ethics committee according to the French Public Health Law (CSP Art L 1121-1.1).

4. There is no evidence of experimental controls including, for example, inoculation of the M-ARCOL system with sterile saliva. It is important to have population controls in the system to identify microorganisms in the samples and not outside.

We thank the reviewer for this suggestion, indeed it would have been the perfect control to inoculate M-ARCOL with sterile saliva, but the matrix effect of the medium is negligible (< 6%) since the volume of saliva injected in the bioreactor is minor compared to the total volume of the bioreactor (2 times 10 ml of saliva in a bioreactor cuve of 300 ml).

5. Was a control carried out to verify that the inoculated bacteria are viable? sequencing identifies microorganisms in the sample regardless of their viability, which could greatly influence the expected results on the system.

We fully agree with the reviewer that the viability of the bacteria was not investigated by plating, that will be implemented in future experiments. Nonetheless, we could envision that the bacteria that adhered to the mucin beads were viable, as most of the gut and oral bacteria were retrieved in the mucosal compartment at any time of our experiment (please refer to Figure 2 and Figure 4).

Literature cited

- Albuquerque-Souza E, Sahingur SE (2022) Periodontitis, chronic liver diseases, and the emerging oral-gut-liver axis. *Periodontol* 2000 89:125–141. <https://doi.org/10.1111/prd.12427>
- Deschamps C, Fournier E, Uriot O, Lajoie F, Verdier C, Comtet-Marre S, Thomas M, Kapel N, Cherbuy C, Alric M, Almeida M, Etienne-Mesmin L, Blanquet-Diot S (2020) Comparative methods for fecal sample storage to preserve gut microbial structure and function in an in vitro model of the human colon. *Appl Microbiol Biotechnol* 104:10233–10247. <https://doi.org/10.1007/s00253-020-10959-4>
- Fournier E, Leveque M, Ruiz P, Ratel J, Durif C, Chalancon S, Amiard F, Edely M, Bezirard V, Gaultier E, Lamas B, Houdeau E, Lagarde F, Engel E, Etienne-Mesmin L, Blanquet-Diot S, Mercier-Bonin M (2023) Microplastics: What happens in the human digestive tract? First evidences in adults using in vitro gut models. *J Hazard Mater* 442:130010. <https://doi.org/10.1016/j.jhazmat.2022.130010>

- Fromentin S, Forslund SK, Chechi K, Aron-Wisniewsky J, Chakaroun R, Nielsen T, Tremaroli V, Ji B, Prifti E, Myridakis A, Chilloux J, Andrikopoulos P, Fan Y, Olanipekun MT, Alves R, Adiouch S, Bar N, Talmor-Barkan Y, Belda E, Caesar R, Coelho LP, Falony G, Fellahi S, Galan P, Galleron N, Helft G, Hoyles L, Isnard R, Le Chatelier E, Julienne H, Olsson L, Pedersen HK, Pons N, Quinquis B, Rouault C, Roume H, Salem J-E, Schmidt TSB, Vieira-Silva S, Li P, Zimmermann-Kogadeeva M, Lewinter C, Søndertoft NB, Hansen TH, Gauguier D, Gøtze JP, Køber L, Kornowski R, Vestergaard H, Hansen T, Zucker J-D, Hercberg S, Letunic I, Bäckhed F, Oppert J-M, Nielsen J, Raes J, Bork P, Stumvoll M, Segal E, Clément K, Dumas M-E, Ehrlich SD, Pedersen O (2022) Microbiome and metabolome features of the cardiometabolic disease spectrum. *Nat Med* 28:303–314. <https://doi.org/10.1038/s41591-022-01688-4>
- Gasmí Benahmed A, Gasmí A, Doşa A, Chirumbolo S, Mujawdiya PK, Aaseth J, Dadar M, Bjørklund G (2021) Association between the gut and oral microbiome with obesity. *Anaerobe* 70:102248. <https://doi.org/10.1016/j.anaerobe.2020.102248>
- Gevers D, Kugathasan S, Denson LA, Vázquez-Baeza Y, Van Treuren W, Ren B, Schwager E, Knights D, Song SJ, Yassour M, Morgan XC, Kostic AD, Luo C, González A, McDonald D, Haberman Y, Walters T, Baker S, Rosh J, Stephens M, Heyman M, Markowitz J, Baldassano R, Griffiths A, Sylvester F, Mack D, Kim S, Crandall W, Hyams J, Huttenhower C, Knight R, Xavier RJ (2014) The treatment-naive microbiome in new-onset Crohn's disease. *Cell Host Microbe* 15:382–392. <https://doi.org/10.1016/j.chom.2014.02.005>
- Hu S, Png E, Gowans M, Ong DEH, de Sessions PF, Song J, Nagarajan N (2021) Ectopic gut colonization: a metagenomic study of the oral and gut microbiome in Crohn's disease. *Gut Pathog* 13:13. <https://doi.org/10.1186/s13099-021-00409-5>
- Iorgulescu G (2009) Saliva between normal and pathological. Important factors in determining systemic and oral health. *J Med Life* 2:303–307
- Kim HJ, Li H, Collins JJ, Ingber DE (2016) Contributions of microbiome and mechanical deformation to intestinal bacterial overgrowth and inflammation in a human gut-on-a-chip. *Proc Natl Acad Sci U S A* 113:E7-15. <https://doi.org/10.1073/pnas.1522193112>
- Kim J, Koo B-K, Knoblich JA (2020) Human organoids: model systems for human biology and medicine. *Nat Rev Mol Cell Biol* 21:571–584. <https://doi.org/10.1038/s41580-020-0259-3>
- Marotz CA, Sanders JG, Zuniga C, Zaramela LS, Knight R, Zengler K (2018) Improving saliva shotgun metagenomics by chemical host DNA depletion. *Microbiome* 6:42. <https://doi.org/10.1186/s40168-018-0426-3>
- Mosca A, Leclerc M, Hugot JP (2016) Gut Microbiota Diversity and Human Diseases: Should We Reintroduce Key Predators in Our Ecosystem? *Front Microbiol* 7:455. <https://doi.org/10.3389/fmicb.2016.00455>
- Park S-Y, Hwang B-O, Lim M, Ok S-H, Lee S-K, Chun K-S, Park K-K, Hu Y, Chung W-Y, Song N-Y (2021) Oral-Gut Microbiome Axis in Gastrointestinal Disease and Cancer. *Cancers (Basel)* 13:2124. <https://doi.org/10.3390/cancers13092124>
- Pham VT, Chassard C, Rifa E, Braegger C, Geirnaert A, Rocha Martin VN, Lacroix C (2019) Lactate Metabolism Is Strongly Modulated by Fecal Inoculum, pH, and Retention Time in PolyFermS Continuous Colonic Fermentation Models Mimicking Young Infant Proximal Colon. *mSystems* 4:e00264-18. <https://doi.org/10.1128/mSystems.00264-18>
- Shah P, Fritz JV, Glaab E, Desai MS, Greenhalgh K, Frachet A, Niegowska M, Estes M, Jäger C, Seguin-Devaux C, Zenhausem F, Wilmes P (2016) A microfluidics-based in vitro model of the gastrointestinal human-microbe interface. *Nat Commun* 7:11535. <https://doi.org/10.1038/ncomms11535>
- Simon-Soro A, Ren Z, Krom BP, Hoogenkamp MA, Cabello-Yeves PJ, Daniel SG, Bittinger K, Tomas I, Koo H, Mira A (2022) Polymicrobial Aggregates in Human Saliva Build the Oral Biofilm. *mBio* 13:e0013122. <https://doi.org/10.1128/mbio.00131-22>
- Van den Abbeele P, Belzer C, Goossens M, Kleerebezem M, De Vos WM, Thas O, De Weirtdt R, Kerckhof F-M, Van de Wiele T (2013) Butyrate-producing Clostridium cluster XIVa species specifically colonize mucins in an in vitro gut model. *ISME J* 7:949–961. <https://doi.org/10.1038/ismej.2012.158>
- Yachida S, Mizutani S, Shiroma H, Shiba S, Nakajima T, Sakamoto T, Watanabe H, Masuda K, Nishimoto Y, Kubo M, Hosoda F, Rokutan H, Matsumoto M, Takamaru H, Yamada M, Matsuda T, Iwasaki M, Yamaji T, Yachida T, Soga T, Kurokawa K, Toyoda A, Ogura Y, Hayashi T, Hatakeyama M, Nakagama H, Saito Y, Fukuda S, Shibata T, Yamada T (2019) Metagenomic and metabolomic analyses reveal distinct stage-specific phenotypes of the gut microbiota in colorectal cancer. *Nat Med* 25:968–976. <https://doi.org/10.1038/s41591-019-0458-7>
- Zihler Berner A, Fuentes S, Dostal A, Payne AN, Vazquez Gutierrez P, Chassard C, Grattepanche F, de Vos WM, Lacroix C (2013) Novel Polyfermentor intestinal model (PolyFermS) for controlled ecological studies: validation and effect of pH. *PLoS One* 8:e77772. <https://doi.org/10.1371/journal.pone.0077772>

March 5, 2023

Dr. Mathieu Almeida
Université Paris-Saclay, INRAE
US 1367 MetaGenoPolis (MGP)
Saclay
France

Re: Spectrum04344-22R1 (In vitro modelling of oral microbial invasion in the human colon)

Dear Dr. Mathieu Almeida:

Your manuscript has been accepted, and I am forwarding it to the ASM Journals Department for publication. You will be notified when your proofs are ready to be viewed.

Sincerely,

Diyen Li
Editor, Microbiology Spectrum
